# Spatiotemporal dynamics of maximal and minimal EEG spectral power

**Melisa Menceloglu[1]¤, Marcia Grabowecky[1,2], Satoru Suzuki[1,2]***

**1** Department of Psychology, Northwestern University, Evanston, IL, United States of America,
**2** Interdepartmental Neuroscience, Northwestern University, Evanston, IL, United States of America

¤ Current address: Cognitive, Linguistic, and Psychological Sciences, Brown University, Providence, RI, United States of America
* satoru@northwestern.edu

**Data Availability Statement:** All analysis data are contained within the paper. The EEG data are available at https://doi.org/10.21985/n2-8hxs-md53.

## Abstract

Oscillatory neural activities are prevalent in the brain with their phase realignment contributing to the coordination of neural communication. Phase realignments may have especially strong (or weak) impact when neural activities are strongly synchronized (or desynchronized) within the interacting populations. We report that the spatiotemporal dynamics of strong regional synchronization measured as maximal EEG spectral power—referred to as *activation*—and strong regional desynchronization measured as minimal EEG spectral power—referred to as *suppression*—are characterized by the spatial segregation of small-scale and large-scale networks. Specifically, small-scale spectral-power activations and suppressions involving only 2–7% (1–4 of 60) of EEG scalp sites were prolonged (relative to stochastic dynamics) and consistently co-localized in a frequency specific manner. For example, the small-scale networks for $\theta$, $\alpha$, $\beta_1$, and $\beta_2$ bands (4–30 Hz) consistently included frontal sites when the eyes were closed, whereas the small-scale network for $\gamma$ band (31–55 Hz) consistently clustered in medial-central-posterior sites whether the eyes were open or closed. Large-scale activations and suppressions involving over 17–30% (10–18 of 60) of EEG sites were also prolonged and generally clustered in regions complementary to where small-scale activations and suppressions clustered. In contrast, intermediate-scale activations and suppressions (involving 7–17% of EEG sites) tended to follow stochastic dynamics and were less consistently localized. These results suggest that strong synchronizations and desynchronizations tend to occur in small-scale and large-scale networks that are spatially segregated and frequency specific. These synchronization networks may broadly segregate the relatively independent and highly cooperative oscillatory processes while phase realignments fine-tune the network configurations based on behavioral demands.

## Introduction

Many studies have investigated macroscopic networks of oscillatory neural activity in humans by examining the spatiotemporal patterns of spectral amplitude, phase, and phase-amplitude relations within and across frequency bands and brain regions using EEG and MEG methods.

**Funding:** National Institutes of Health T32 NS047987 to MM.

**Competing interests:** The authors have declared that no competing interests exist.

Those studies typically examined oscillatory interactions in specific regions of interest or characterized networks of oscillatory activities and their connectivity by analyzing the structures of correlation matrices (derived from pairwise temporal associations of spectral amplitude and/or phase across space and/or frequencies), often utilizing clustering methods and/or graph theoretic measures derived from correlation matrices [1–9; see 10 for a review]. These approaches have productively characterized frequency-specific interactions in targeted regions as well as features of frequency-specific networks that correlate with a variety of behavioral functions [11–23] and mental states [24–27]. Nevertheless, for characterizing networks, correlation-matrix based approaches (utilizing pairwise temporal associations) trade time dimension for revealing (static, time-averaged) spatial connectivity structures. As a complementary approach, the current study investigated general rules governing the spatiotemporal dynamics of EEG spectral power.

EEG recorded at each scalp electrode is thought to reflect the macroscopically summed field potentials arising from the current sources/sinks generated by the regional population of large cortical pyramidal cells that are aligned in parallel and perpendicular to the cortical surface [e.g., 28]. Thus, it is reasonable to infer that larger spectral power at a scalp site reflects more extensive oscillatory synchronization within the accessible regional pyramidal-cell population (resulting in less cancellation of field oscillations), whereas smaller spectral power reflects less extensive oscillatory synchronization which may result from less cells being engaged in oscillatory activity and/or desynchronization of oscillations (resulting in greater cancellation of field oscillations). Thus, spectral power at each scalp electrode reflects the degree of oscillatory synchronization of the regional population of the contributing cortical pyramidal cells (though non-oscillatory components and some artifacts may also contribute to spectral power; see Caveats). Our spatial resolution was 1–2 cm after surface-Laplacian transforming the scalp-recorded EEG to infer the macroscopic current-source/sink densities at the electrode sites (see Materials and methods; also referred to as dura potential; e.g., [28]).

Phenomenologically, we became intrigued by the observation that spectral power sometimes spontaneously maximized or minimized in isolation exclusively at a single site, whereas at other times, spectral power globally maximized or minimized over a large number of sites. For convenience, we refer to maximal spectral power (defined by an upper percentile threshold; see Results and discussion) as *activation* in the sense of activation of extensive regional oscillatory synchronization, and minimal spectral power (defined by a lower percentile threshold) as *suppression* in the sense of suppression of regional oscillatory synchronization.

Are there general rules governing the spontaneous fluctuations in the spatial extent and clustering of spectral-power activations and suppressions? Our strategy was to examine the number of concurrently activated or suppressed sites as a function of time. In particular, we determined whether consistent spatial patterns of activations and suppressions emerged as a function of the number of concurrently activated or suppressed sites. For example, if a specific group of *n* sites formed a network, that is, if a specific group of *n* sites tended to be concurrently activated or suppressed, whenever the number of concurrently activated or suppressed sites happened to be *n*, a specific group of sites should be consistently included with elevated probability.

This approach may appear similar to finding clusters in a correlation matrix (e.g., constructed from binarized values, +1 for activation and –1 for suppression), but there are some crucial differences. First, the current method allows direct examinations of the spatial consistency and temporal dynamics of the clusters of activations and suppressions of different sizes that actually occur (rather than inferring time-averaged clusters from correlation matrices by using cluster number as a fitting parameter). For instance, suppose activations at site A were variously correlated with activations at other sites, but rare isolated activations consistently

occurred at site A. Analyses of correlation matrices would miss it (because pairwise correlations do not track instantaneous spatial patterns), but the current analysis would detect it. The current analysis would also reveal the dynamics (e.g., average duration) of such rare but consistently isolated activations occurring at site A. Second, the current method allows examinations of how temporal contexts influence the clustering of activations and suppressions. For instance, the composition of a small cluster of activations may differ depending on whether it occurs in the midst of a persisting period of small-cluster activations, immediately following a period of widespread activations, or immediately preceding an emergence of widespread activations. Information about temporal contexts is unavailable in correlation matrices. Overall, the current analysis is complementary to structural analyses applied to correlation matrices.

To increase the generalizability of our results, we examined spontaneous spatiotemporal fluctuations in spectral-power activations and suppressions while participants rested with their eyes closed, rested with their eyes open in a dark room, or casually viewed a silent nature video.

The results provided converging evidence suggesting that the spatiotemporal dynamics of spectral-power activations and suppressions are characterized by the spatial segregation of small-scale and large-scale networks. Specifically, small-scale spectral-power activations and suppressions involving only 2–7% (1–4 of 60) of EEG scalp sites were prolonged (relative to stochastic dynamics), consistently co-localized in a frequency specific manner, and were stable while the spatial extent of activations/suppressions remained in the small-scale range. Large-scale activations and suppressions involving over 17–30% (over 10–18 of 60) of EEG sites were also prolonged, generally clustered in regions complementary to where small-scale activations and suppressions clustered, and were stable while the spatial extent of activations/suppressions remained in the large-scale range. These macroscopic networks of strong synchronization and desynchronization may broadly segregate the relatively independent and highly cooperative oscillatory processes while phase realignments may fine-tune the network configurations based on behavioral demands.

## Materials and methods

### Participants

Fifty-two Northwestern University students (35 women, 1 non-binary who declined to identify their gender as either woman or man; mean age of 20.8 years, ranging from 18 to 29 years, standard deviation of 2.5 years) signed a written consent form to participate for monetary compensation ($10/hr). All were right-handed, had normal hearing and normal or corrected-to-normal vision, and had no history of concussion. They were tested individually in a dimly lit or dark room. The study protocol was approved by the Northwestern University Institutional Review Board. Participants p1-p7 and p12-p28 ($N = 24$) participated in a rest-with-eyes-closed condition in which EEG was recorded for ~5 min while participants rested with their eyes closed and freely engaged in spontaneous thoughts. Participants p8-p28 ($N = 21$) subsequently participated in a silent-nature-video condition in which EEG was recorded for ~5 min while they viewed a silent nature video. To evaluate test-retest reliability, the silent-nature-video condition was run twice (20–30 min apart), labeled as earlier viewing and later viewing in the analyses. A generic nature video was presented on a 13-inch, 2017 MacBook Pro, 2880 (H)-by-1800(V)-pixel-resolution LCD monitor with normal brightness and contrast settings, placed 100 cm away from participants, subtending approximately 16˚(H)-by-10˚(V) of visual angle. Participants p29-p52 ($N = 24$) participated in the replication of the rest-with-eyes-closed condition and subsequently participated in a rest-with-eyes-open-in-dark condition which was the same as the former except that the room was darkened and participants kept their eyes

open while blinking naturally. Subsets of these EEG data were previously analyzed for a different purpose [29–31].

## EEG recording and pre-processing

While participants rested with their eyes closed, rested with their eyes open in dark, or viewed a silent nature video for approximately 5 min, EEG was recorded from 64 scalp electrodes (although we used a 64-electrode montage, we excluded signals from noise-prone electrodes, *Fpz*, *Iz*, *T9*, and *T10*, from analyses) at a sampling rate of 512 Hz using a BioSemi ActiveTwo system (see www.biosemi.com for details). Electrooculographic (EOG) activity was monitored using four face electrodes, one placed lateral to each eye and one placed beneath each eye. Two additional electrodes were placed on the left and right mastoid area. The EEG data were pre-processed using EEGLAB and ERPLAB toolboxes for MATLAB [32, 33]. The data were re-referenced offline to the average of the two mastoid electrodes, bandpass-filtered at 0.01 Hz-80 Hz, and notch-filtered at 60 Hz (to remove power-line noise that affected the EEG signals from some participants). For the EEG signals recorded while participants rested with the eyes open in dark or while they viewed a silent nature video, an Independent Component Analysis (ICA) was conducted using EEGLABs' *runica* function [34, 35]. Blink related components were visually identified (apparent based on characteristic topography) and removed (no more than two components were removed per participant).

We surface-Laplacian transformed all EEG data for the following reasons. The transform (1) substantially reduces volume conduction (reducing the electrode-distance dependent component of phase alignment to within adjacent sites, or 1–2 cm, for a 64-channel montage [36]), (2) virtually eliminates the effects of reference electrode choices (theoretically; we also verified this for our data), and (3) provides a data-driven method to fairly accurately map scalp-recorded EEG to macroscopic current-source/sink densities over the cortical surface [37–39]. We used Perrin and colleagues' algorithm [40–42] with the "smoothness" value, $\lambda = 10^{-5}$ (recommended for 64 channels [36]). We refer to the surface-Laplacian transformed EEG signals that represent the macroscopic current source/sink densities under the 60 scalp sites (with the 4 noise-prone sites removed from analyses; see above) simply as EEG signals. These EEG-recording and pre-processing procedures were identical to those used in our prior study [29].

**EEG analysis.** We used the temporal derivative of EEG as in our prior studies that examined all [29] or a subset [31] of the current EEG data for different purposes. While the rationale for taking the temporal derivative of EEG is detailed in [29], it offers the following advantages. First, EEG temporal derivatives may highlight oscillatory dynamics by reducing the non-oscillatory $1/f^{\beta}$ spectral backgrounds when $\beta\sim1$, which was the case for our EEG data on the timescale of several seconds [29]. Second, EEG temporal derivatives may be considered a "deeper" measure of neural activity than EEG in the sense that scalp-recorded potentials are caused by the underlying neural currents and taking EEG temporal derivative macroscopically estimates those currents (as currents in RC circuits are proportional to the temporal derivative of the corresponding potentials). Third, EEG temporal derivatives are drift free. Prior studies used EEG temporal derivatives for similar reasons [e.g., 43–45], providing some evidence suggesting that EEG temporal derivatives yield more effective neural features than EEG for brain-computer interface [45].

We generated phase-scrambled controls whose spectral power fluctuated stochastically while maintaining the time-averaged spectral-amplitude profiles of the actual EEG data. While phase-scrambling can be performed using several different methods, we chose discrete cosine transform, DCT [46]. In short, we transformed each ~5 min EEG waveform with type-2 DCT, randomly shuffled the signs of the coefficients, and then inverse-transformed it with type-3

DCT (the "inverse DCT"), which yielded a phase-scrambled version. DCT phase-scrambling is similar to DFT (discrete Fourier transform) phase-scrambling except that it is less susceptible to edge effects. We previously verified that DCT phase-scrambling generated waveforms whose spectral-power fluctuations conformed to exponential distributions [29, Fig 2] indicative of a Poisson point process (an unpredictable and memory-free process), with negligible distortions to the time-averaged spectral-amplitude profiles of the actual EEG data [29, Fig 1].

To investigate how spectral power (amplitude squared of sinusoidal components) fluctuated over time, we used a Morlet wavelet-convolution method suitable for time-frequency decomposition of signals containing multiple oscillatory sources of different frequencies (see [36] for a review of different methods for time-frequency decomposition). Each Morlet wavelet is a Gaussian-windowed sinusoidal templet characterized by its center frequency as well as its temporal and spectral widths that limit its temporal and spectral resolution. We decomposed each EEG waveform (i.e., its temporal derivative) into a time series of spectral power using Morlet wavelets with 200 center frequencies, $f_c$'s, between 3 Hz and 60 Hz. The $f_c$'s were logarithmically spaced because neural temporal-frequency tunings tend to be approximately logarithmically scaled [47, 48]. The accompanying $n$ factors (roughly the number of cycles per wavelet, $n = 2\pi f \cdot SD$, where $SD$ is the wavelet standard deviation) were also logarithmically spaced between 3 and 16, yielding temporal resolutions ranging from $SD = 159$ ms (at 3 Hz) to $SD = 42$ ms (at 60 Hz) and spectral resolutions of $FWHM$ (full width at half maximum) = 2.36 Hz (at 3 Hz) to $FWHM = 8.83$ Hz (at 60 Hz). These values struck a good balance for the time-frequency trade-off, and are typically reported in the literature [36]. The spectral-power values were then averaged within the commonly considered frequency bands, $\theta$ (4–7 Hz), $\alpha$ (8–12 Hz), $\beta$ (13–30 Hz), and $\gamma$ (31–55 Hz). For some analyses, we subdivided the $\beta$ band into lower, $\beta_1$ (13–20 Hz), and higher, $\beta_2$ (21–30 Hz), portions.

## Results and discussion

### Setting thresholds to define maximal and minimal spectral power

The goal of this study was to uncover rules (if any) that govern the spatiotemporal dynamics of maximal and minimal spectral power. To this end, we first operationalized maximal and minimal spectral power as the top-$i$th-percentile and the bottom-$j$th-percentile spectral power, which we refer to as activation and suppression (see Introduction).

The number of concurrently activated/suppressed sites dynamically increases and decreases, making the spatial extent of spectral-power activations/suppressions dynamically expand and contract. In this context, focal activations/suppressions occurring exclusively at single sites are special in that they may reflect the instances of relatively independent controls of activation/suppression. Thus, as a first step, we calibrated activation and suppression thresholds so that the instances of focal activation and focal suppression were detected with maximal sensitivity. Specifically, we determined the percentile thresholds (applied per scalp site per frequency band per condition per participant) that maximized the prevalence of focal activations and suppressions relative to chance levels.

For a broad range of thresholds, the instances of focal activations (the upper row in Fig 1) and focal suppressions (the lower row in Fig 1) were elevated relative to the corresponding Binomial probabilities given by $60 \cdot p(1-p)^{60-1}$, where $p$ is the percentile threshold and 60 is the number of sites. Notably, the 8th-percentile threshold clearly maximized the instances of both focal activations and suppressions for all frequency bands across all conditions (Fig 1). We thus defined activation as yielding the top-8th-percentile spectral power and suppression as yielding the bottom-8th-percentile spectral power (per scalp site per frequency band per

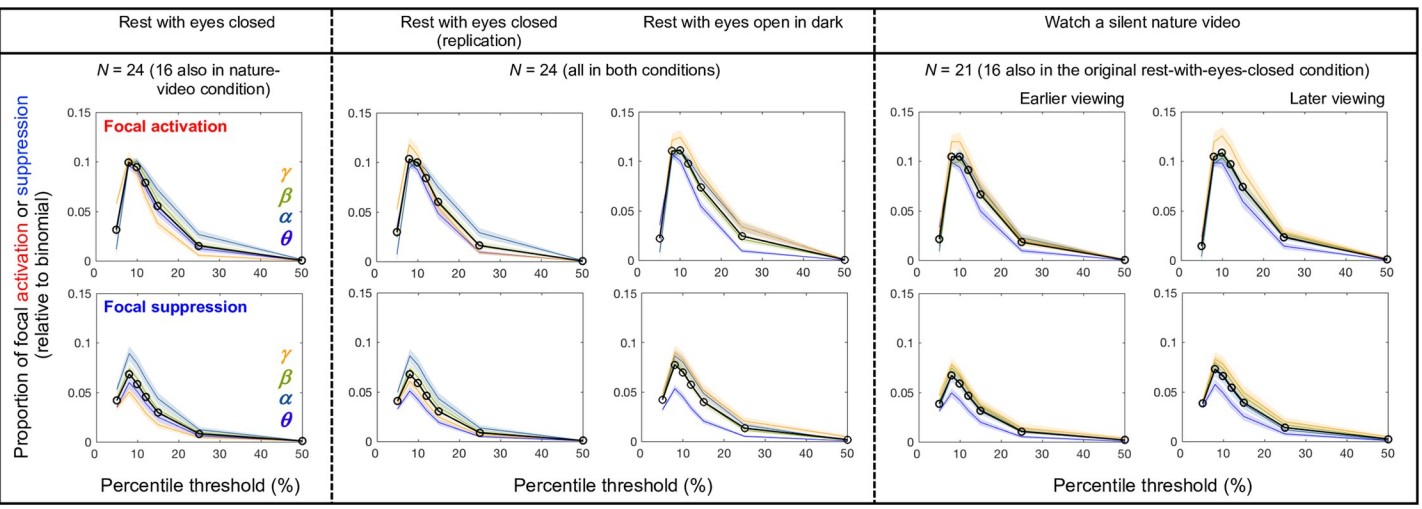

**Fig 1. Proportions of focal activations (top-percentile spectral power occurring exclusively at only one site; top row) and focal suppressions (bottom-percentile spectral power occurring exclusively at only one site; bottom row) as a function of percentile threshold (relative to binomial chance levels).** The cooler and warmer colors indicate the lower and higher frequency bands, $\theta$ (4–7 Hz), $\alpha$ (8–12 Hz), $\beta$ (13–30 Hz) and $\gamma$ (31–55 Hz). The black lines with open circles show averages across the frequency bands. The columns correspond to the five behavioral conditions. For all frequency bands across all conditions, the instances of focal activations and focal suppressions were consistently maximized with the 8$^{th}$-percentile threshold. We thus defined activation as yielding the top 8% spectral power and suppression as yielding the bottom 8% spectral power.

condition per participant). As expected, all phase-scrambled controls conformed to the Binomial probabilities (not shown, but they would yield straight horizontal lines at $y = 0$ in Fig 1).

## Dynamics of the spatial scale of spectral-power activations and suppressions

To examine the dynamics of the spatial scale of spectral-power activations and suppressions, we examined the time series of the number of activated or suppressed sites, $n_{sites}$. Increases (or decreases) in $n_{sites}$ indicate expansions (or contractions) in the spatial scale of activations or suppressions. An example time series of $n_{sites}$ (from 50 sec to 100 sec) for activations and suppressions for $\alpha$ power for one participant in the rest-with-eyes-closed condition are shown in Fig 2A–2D. The instances of small-scale (involving 1–4 sites) and large-scale (involving 10 + sites) activations and suppressions are highlighted (in red for activations and blue for suppressions). Note that we assigned these small-scale and large-scale designations based on the fact (see below) that activations and suppressions involving 1–4 sites and those involving 10 + sites exhibited distinct spatial distributions.

Even from the illustrative example shown in Fig 2A–2D it is clear that the typical intervals of large-scale activations (the red highlighted portions above $n_{sites} = 10$) were substantially longer for the actual data (Fig 2A) than for the phase-scrambled control (Fig 2B). Though less apparent, one can also see that the typical intervals of small-scale activations (the red highlighted portions below $n_{sites} = 4$) were longer for the actual data (Fig 2A) than for the phase-scrambled control (Fig 2B). These observations also apply to suppression. The typical intervals of large-scale suppressions (the blue highlighted sections above $n_{sites} = 10$) were substantially longer for the actual data (Fig 2C) than for the phase-scrambled control (Fig 2D), and the typical intervals of small-scale suppressions (the blue highlighted portions below $n_{sites} = 4$) were also longer for the actual data (Fig 2C) than for the phase-scrambled control (Fig 2D), albeit less apparent.

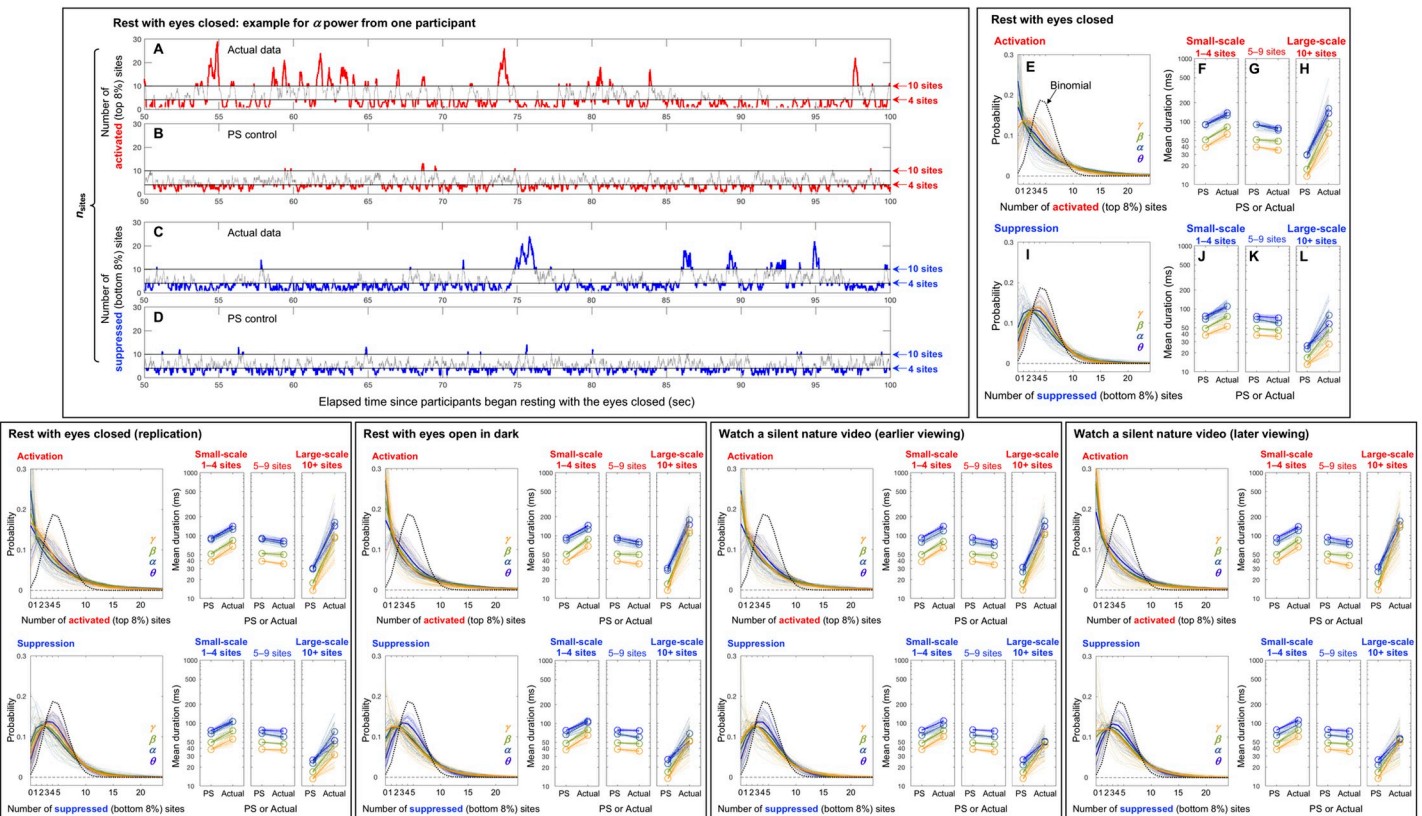

**Fig 2. Elevated instances of focal (involving 1–2 sites) and large-scale (involving 10+ sites) activations and suppressions, as well as extended average durations of small-scale (involving 1–4 sites) and large-scale (involving 10+ sites) activations and suppressions.** Activation is defined as spectral power in the top 8% and suppression is defined as spectral power in the bottom 8% (see main text and Fig 1). A-D. An example of the temporal fluctuations of the number of concurrently activated or suppressed sites for a representative participant for $\alpha$ power. The instances of small-scale (involving 1–4 sites) and large-scale (involving 10+ sites) activations and suppressions are highlighted. A. The number of activated sites as a function of time. B. The same as A but for the phase-scrambled control. C. The number of suppressed sites as a function of time. D. The same as C but for the phase-scrambled control. E-L. The probabilities and average durations of small-scale (involving 1–4 sites), intermediate-scale (involving 5–9 sites), and large-scale (involving 10+ sites) activations and suppressions for the rest-with-eyes-closed condition. The cooler and warmer colors indicate the lower and higher frequency bands, $\theta$ (4–7 Hz), $\alpha$ (8–12 Hz), $\beta$ (13–30 Hz) and $\gamma$ (31–55 Hz). The solid lines indicate the means while the faint lines show the data from individual participants. E. The probabilities of concurrent activations of different numbers of sites. The thick black dotted curve shows the chance (binomial) distribution. Only the actual data are shown because the phase-scrambled controls conformed to the binomial distribution. Note that the instances of focal (involving 1–2 sites) and large-scale (involving 10+ sites) activations were elevated relative to chance. F-H. The average durations of small-scale (F), intermediate-scale (G), and large-scale (H) activations. Note that the average durations were extended for small- and large-scale (but not intermediate-scale) activations relative to their phase-scrambled controls. I and J-L. The same as E and F-H, but for suppression, showing the same pattern as activation. The remaining panels in the bottom row parallel E-L, showing the results for the four remaining conditions, which are all virtually identical.

Average durations of small-scale ($n_{\text{sites}} \leq 4$, involving 1–4 sites), large-scale ($n_{\text{sites}} \geq 10$, involving 10+ sites), and intermediate-scale ($5 \leq n_{\text{sites}} \leq 9$, involving 5–9 sites) activations and suppressions as well as the probability distributions of $n_{\text{sites}}$ for the rest-with-eyes-closed condition are summarized for the four representative frequency bands ($\theta$, $\alpha$, $\beta$, and $\gamma$) in Fig 2E–2L. For all frequency bands, the probabilities of $n_{\text{sites}} \leq 2$ and $n_{\text{sites}} \geq 10$ were elevated for both activations (Fig 2E) and suppressions (Fig 2I) relative to the binomial predictions (the black dotted curves in Fig 2E and 2I: $P_{Binom}(n_{sites}) = \frac{N!}{(N-n_{sites})! n_{sites}!} p^{n_{sites}} (1-p)^{N-n_{sites}}$, where $N = 60$ is the total number of sites, $p = 0.08$ is the threshold percentile, and $n_{\text{sites}}$ is the number of concurrently activated or suppressed sites by chance (the data for the phase-scrambled controls conformed to the Binomial predictions; not shown). These results suggest that focal ($n_{\text{sites}} \leq 2$) and large-scale ($n_{\text{sites}} \geq 10$) activations and suppressions are actively maintained.

Consistent with our observations above (Fig 2A–2D), the average durations of small-scale ($n_{\text{sites}} \leq 4$) activations (Fig 2F), small-scale suppressions (Fig 2J), large-scale ($n_{\text{sites}} \geq 10$) activations (Fig 2H), and large-scale suppressions (Fig 2L) were all extended relative to their phase-scrambled controls (actually, three phase-scrambled controls were averaged for greater reliability) for all frequency bands and for all participants (faint lines). In contrast, the average durations of intermediate-scale ($5 \leq n_{\text{sites}} \leq 9$) activations (Fig 2G) and suppressions (Fig 2K) were equivalent to their phase-scrambled controls. We note that the average-duration results should be interpreted with the caveat that higher- (or lower-) probability events tend to yield longer (or shorter) average durations even for stochastic variations. Nevertheless, the average-duration results still provide unique information in that a higher temporal probability of activations/suppressions does not necessarily imply longer average durations because a higher temporal probability may result from an increased frequency of unextended activation/suppression intervals. We also note that the average durations were generally shorter for higher-frequency bands for both the actual data and their phase-scrambled controls. This reflects the fact that the temporal resolutions of Morlet wavelets were set to be higher for the wavelets with higher center frequencies to achieve a reasonable time-frequency trade-off (see Materials and methods). Nevertheless, the approximately parallel lines seen in Fig 2F–2H and 2J–2L indicate that the average durations of small-scale and large-scale activations and suppressions were extended (relative to their phase-scrambled controls) by similar factors (note the logarithmic scale) regardless of frequency band.

All these results characterizing the dynamics of $n_{\text{sites}}$ in the rest-with-eyes-closed condition replicated in the remaining behavioral conditions: the replication of the rest-with-eyes-closed condition, the rest-with-eyes-open-in-dark condition, and the earlier and later instances of the silent-nature-video condition (the lower four panels in Fig 2).

The results so far have shown that the occurrences of focal (involving 1–2 sites) and large-scale (involving 10+ sites) activations and suppressions were increased and that the average durations of small-scale (involving 1–4 sites) and large-scale (involving 10+ sites) activations and suppressions were extended relative to stochastic dynamics while the average durations of intermediate-scale (involving 5–9 sites) activations and suppressions were equivalent to stochastic dynamics (though the average duration results need to be interpreted with a caveat; see above). These results, consistent across all frequency bands, behavioral conditions, and participants, suggest that the brain actively maintains relatively isolated small-scale networks (involving 1–4 sites) and highly cooperative large-scale networks (involving 10+ sites) of spectral-power activations and suppressions while intermediate-scale activations and suppressions tend to occur stochastically in transit.

## Spatial distributions of spectral-power activations and suppressions

The next question we asked was whether small-scale (involving 1–4 sites) and large-scale (involving 10+ sites) activations and suppressions clustered in specific regions. In other words, were distinct neural populations preferentially involved in relatively isolated versus highly coordinated spectral-power activations and suppressions? To address this question, we examined the spatial probability distribution of activations and suppressions as a function of $n_{\text{sites}}$ (the number of concurrently activated or suppressed sites) relative to the chance probability of 1/60 (because we used 60 sites).

The spatial probability distributions are shown for each frequency band in Fig 3 ($\theta$), Fig 4 ($\alpha$), Fig 5 ($\beta_1$), Fig 6 ($\beta_2$), and Fig 7 ($\gamma$). We split $\beta$ band (13–30 Hz) into the lower $\beta_1$ (13–20 Hz) and upper $\beta_2$ (21–30 Hz) sub-bands because they yielded distinct spatial probability distributions. In each figure, the rows of spatial-probability-distribution plots are organized by

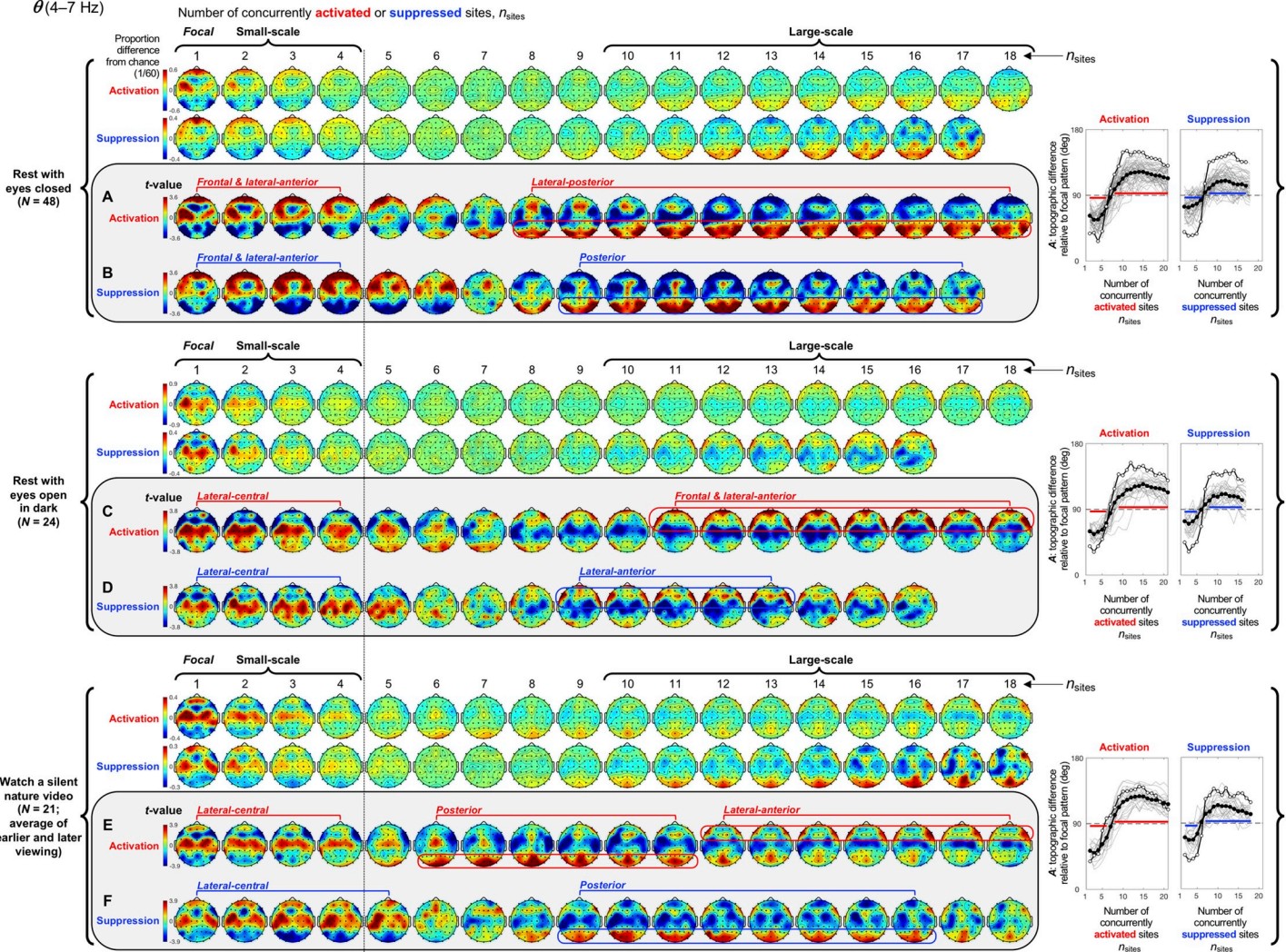

**Fig 3. Spatial-probability distributions of spectral-power activations and suppressions as a function of the number of concurrently activated/suppressed sites,** $n_{sites}$**, for the** $\theta$ **band (4–7 Hz).** Activation and suppression are defined as spectral power in the top and bottom 8% (see main text and Fig 1). The designations of small-scale (involving 1–4 sites) and large-scale (involving 10+ sites) clusters are based on their topographic complementarity (shown here) and their extended average durations (Fig 2). The major row divisions correspond to the three behavioral conditions. For each condition, the spatial-probability-distribution plots are presented in four rows (row 1-row 4). Rows 1 and 2 show the spatial-probability distributions of activations and suppressions, respectively, for increasing numbers of concurrently activated sites, $n_{sites}$. The above- and below-chance occurrences of activations/suppressions are color-coded with warmer and cooler colors, with the green color indicating the chance level. The values indicate the proportions of deviations from the chance level (1/60); for example, +0.3 indicates 30% more frequent occurrences than expected by chance, whereas –0.6 indicates 60% less frequent occurrences than expected by chance. Rows 3 and 4 are identical to rows 1 and 2 except that they present the inter-participant consistency of deviations from the chance level as $t$-values with the critical values for Bonferroni-corrected statistical significance ($\alpha$ = 0.05, 2-tailed) shown on the color bars. We focused on these $t$-value plots (highlighted and labeled **A-F**) for making inferences (see main text). In most cases, all participants yielded concurrent activations and suppressions involving up to 18 sites, but in some cases some participants yielded concurrent activations or suppressions involving fewer sites. In those cases, the spatial-probability-distribution plots are presented up to the maximum number of concurrently activated or suppressed sites to which all participants contributed. One can see that small-scale activations and suppressions (columns 1–4) occurred in specific regions while large-scale activations and suppressions (columns 10+) generally occurred in the complementary regions (**A-F**). This spatial complementarity between small- and large-scale activations and suppressions is quantified in the accompanying line graphs on the right, plotting $A$, the vector angles (in degrees) between the spatial distribution of focal activations/suppressions and the spatial distributions of multi-site activations/suppressions involving increasing numbers of sites (see main text). $A < 90°$ indicates that multi-site activation/suppression distributions were spatially similar to focal activation/suppression distributions (with $A = 0°$ indicating that they were identical up to a scalar factor), $A = 90°$ indicates that multi-site distributions were orthogonal (unrelated) to focal distributions, and $A > 90°$ indicate that multi-site distributions were increasingly spatially complementary to focal distributions (with $A = 180°$ indicating a complete red-blue reversal). The thick lines with filled circles indicate the mean angles and the gray lines show the data from individual participants. Bonferroni-corrected statistical significance ($\alpha$ = 0.05, 2-tailed) for the negative and positive deviations from $A = 90°$ are indicated with the horizontal lines just below and above the dashed line indicating 90°. The lines with open circles show the vector angles computed from the participant-averaged spatial-probability distributions.

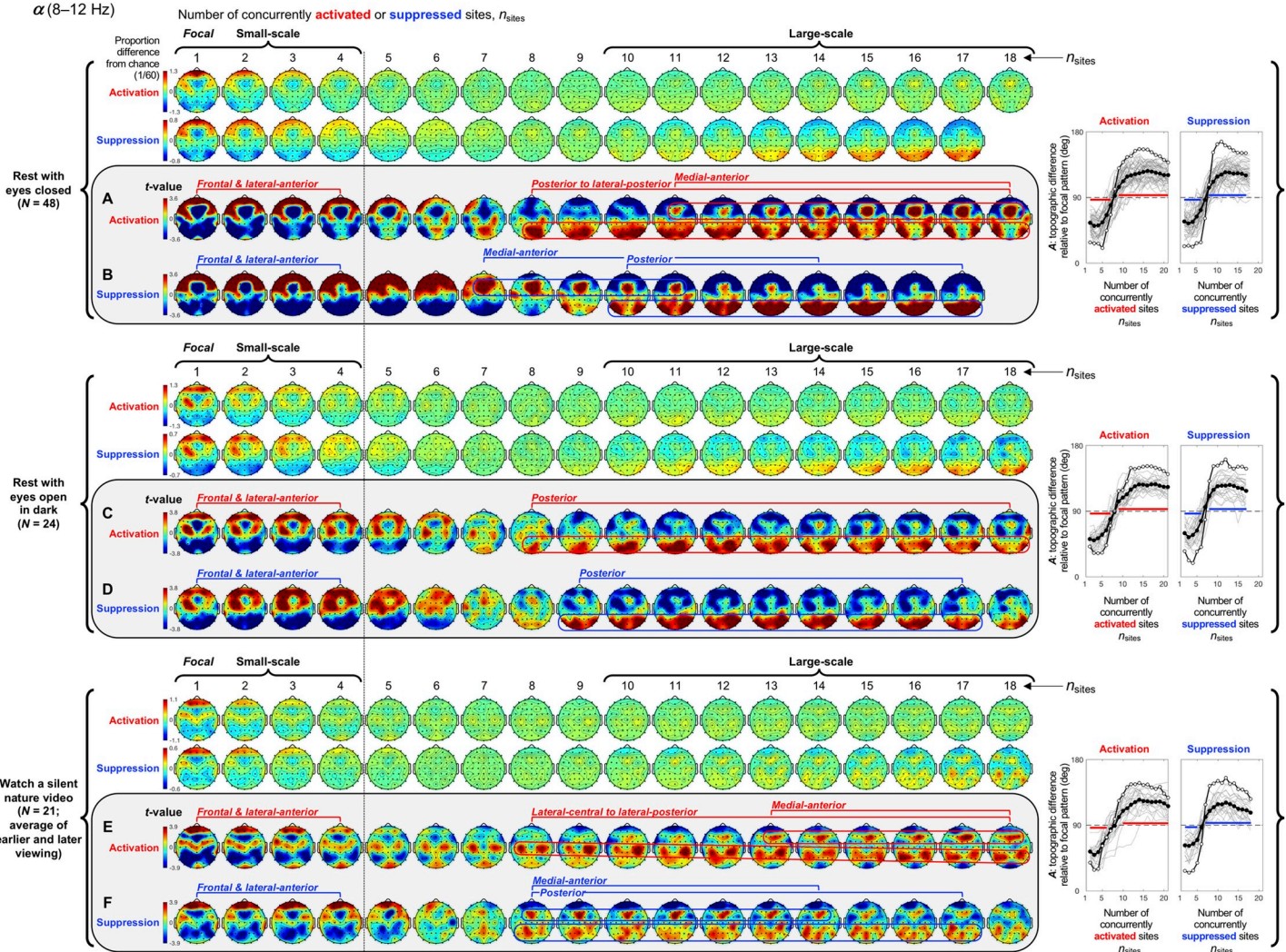

**Fig 4. Spatial-probability distributions of spectral-power activations and suppressions as a function of the number of activated/suppressed sites, $n_{\text{sites}}$, for the $\alpha$ band (8–12 Hz).** The formatting is the same as in Fig 3.

three behavioral conditions, resting with eyes closed (with the original and replication data combined), resting with eyes open in dark, and watching a silent nature video (with the earlier and later viewing data averaged). For each condition, the top two rows show the spatial probability distributions of activations (the upper row) and suppressions (the lower row) for increasing values of $n_{\text{sites}}$, measured as the proportion difference from the chance level (1/60). For instance, a value of 0.4 indicates 40% more frequent occurrences of activations or suppressions than expected by chance; a value of –0.8 indicates 80% less frequent occurrences of activations or suppressions than expected by chance. Although this provides a reasonable measure of positive and negative deviations from the chance level, the positive deviations are necessarily diluted for larger values of $n_{\text{sites}}$ (capped at $1/n_{sites} - 1/60$ because, for example, if triple activation, $n_{\text{sites}} = 3$, always occurred at the same group of three sites, the spatial probability of activation at each of the three sites would be 1/3 and 0 elsewhere) and negative deviations are floored at –1 (cannot be lower than 100% below the chance level). We thus made our inferences based on the robustness of the positive and negative deviations from the chance level

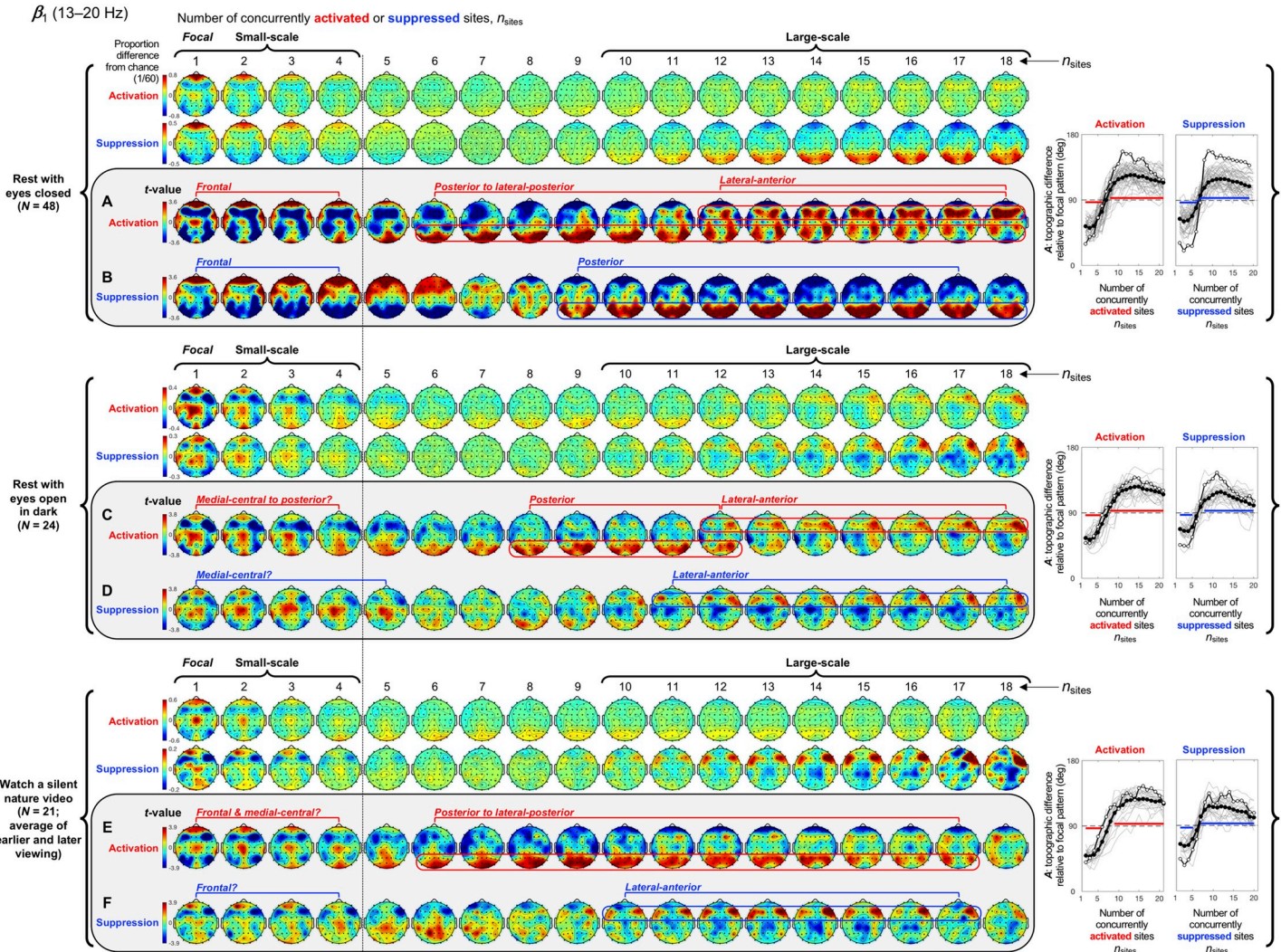

**Fig 5. Spatial-probability distributions of spectral-power activations and suppressions as a function of the number of activated/suppressed sites, $n_{sites}$, for the $\beta_1$ band (13–20 Hz).** The formatting is the same as in Figs 3 and 4.

computed as *t*-values (the mean probability difference from the chance level divided by the standard error with participants as the random effect). Spatial probability distributions based on *t*-values are shown in the bottom two rows for each behavioral condition (highlighted with gray-shaded rectangles and labeled A-F in Figs 3–7). The values required for Bonferroni-corrected significant deviations from the chance level ($\alpha = 0.05$, 2-tailed, accounting for comparisons at 60 sites) are indicated on the color bars. Cooler colors indicate below-chance probabilities, warmer colors indicate above-chance probabilities, and the green color indicates the chance level.

**Spatial distributions of small-scale spectral-power activations and suppressions.**
Small-scale (involving 1–4 sites) activations and suppressions consistently clustered in overlapping and frequency-specific regions in each condition. For the $\theta$ band, small-scale activations and suppressions clustered in the frontal and lateral-anterior regions in the eyes-closed condition (Fig 3A and 3B, left columns labeled "Small-scale"), whereas they clustered in the lateral-central regions in the eyes-open conditions (Fig 3C–3F). For the $\alpha$ band, small-scale

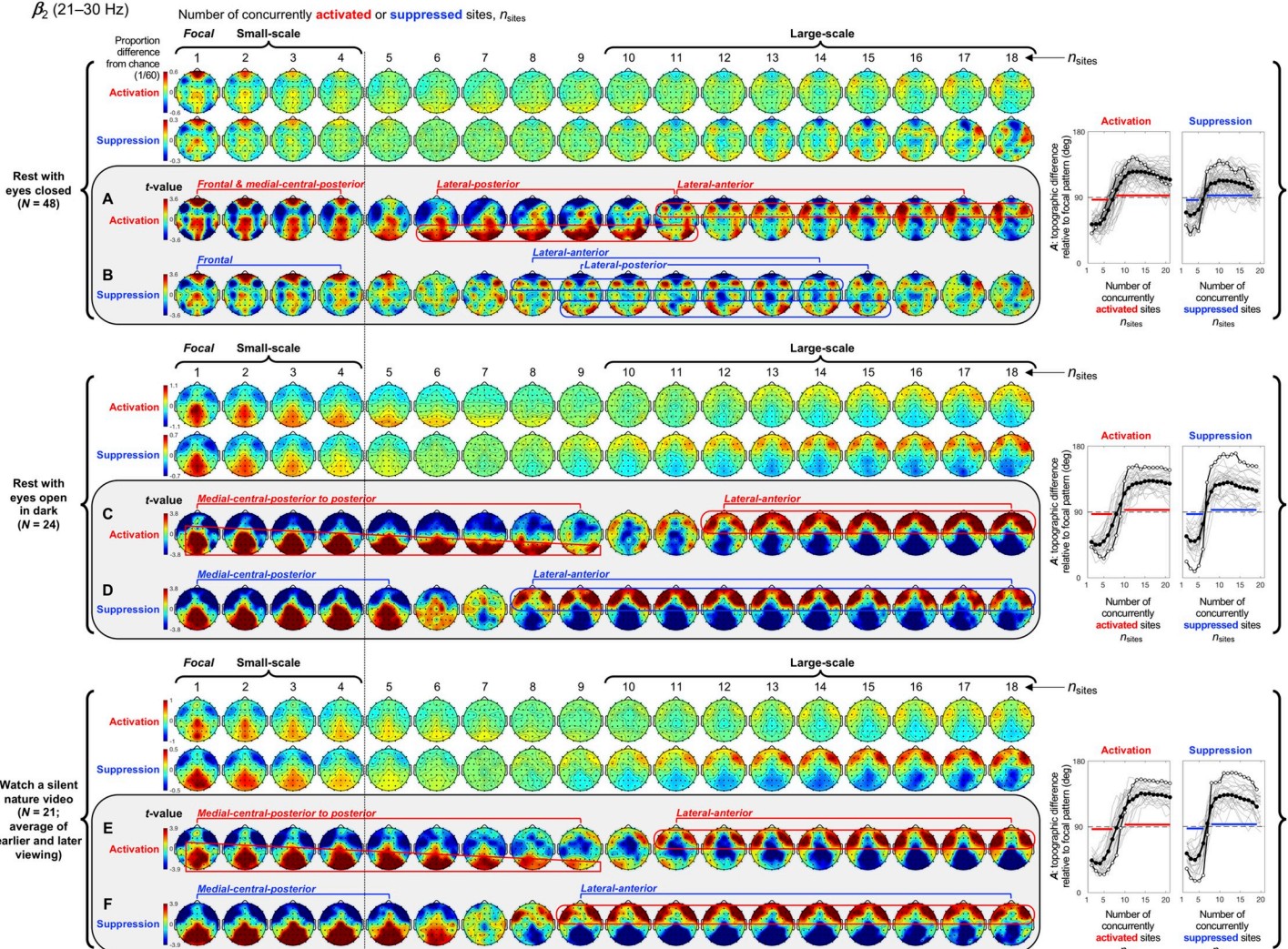

**Fig 6. Spatial-probability distributions of spectral-power activations and suppressions as a function of the number of activated/suppressed sites, $n_{sites}$, for the $\beta_2$ band (21–30 Hz).** The formatting is the same as in Figs 3–5.

activations and suppressions clustered in the frontal and lateral-anterior regions largely across all conditions (Fig 4A–4F). For the $\beta_1$ band, small-scale activations and suppressions clustered in the frontal regions in the eyes-closed condition (Fig 5A and 5B), whereas they only weakly clustered in the frontal, medial-central, and/or posterior regions in the eyes-open conditions (Fig 5C–5F). For the $\beta_2$ band, small-scale activations and suppressions primarily clustered in the frontal regions (potentially also in the medial-central-posterior regions) in the eyes-closed condition (Fig 6A and 6B), whereas they clustered in the medial-central-posterior regions in the eyes-open conditions (Fig 6C–6F). For the $\gamma$ band, small-scale activations and suppressions consistently clustered in the medial-central-posterior regions across all conditions (Fig 7A–7F).

Overall, small-scale activations and suppressions clustered similarly across all conditions for the $\gamma$ band (in the medial-central-posterior regions; Fig 7A–7F, left columns labeled "Small-scale") and also for the $\alpha$ band (in the frontal and lateral-anterior regions; Fig 4A–4F,) to some degree. In the eyes-closed condition, small-scale activations and suppressions for the

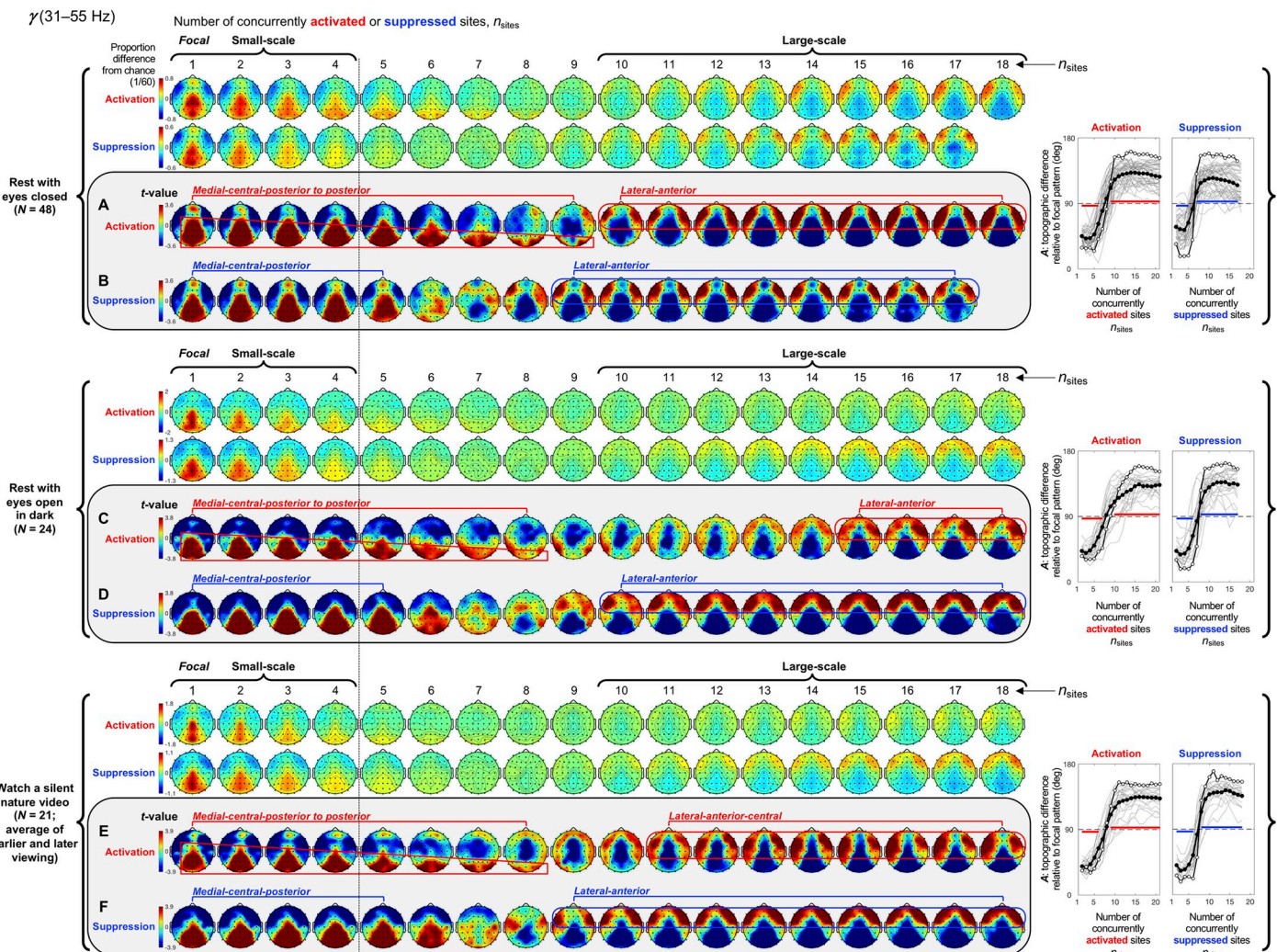

**Fig 7. Spatial-probability distributions of spectral-power activations and suppressions as a function of the number of activated/suppressed sites, $n_{sites}$, for the $\gamma$ band (31–55 Hz).** The formatting is the same as in Figs 3–6.

$\theta$, $\alpha$, $\beta_1$, and $\beta_2$ bands consistently included the frontal sites (Figs 3–6, parts A-B). Interestingly, small-scale activations and suppressions for the $\beta_2$ band clustered similarly to the $\beta_1$ band in the eyes-closed condition (prominently in the frontal regions; compare Figs 6A, 6B with 5A, 5B), whereas they clustered similarly to the $\gamma$ band in the eyes-open conditions (in the medial-central-posterior regions; compare Figs 6C–6F with 7C–7F). Further, for the $\beta_2$ band in the eyes-open conditions and the $\gamma$ band in all conditions, focal-to-intermediate-scale activations systematically shifted from the medial-central-posterior regions to the posterior regions as the number of activated sites increased from 1 to about 9 (highlighted with red trapezoids in Fig 6C and 6E, and Fig 7A, 7C and 7E), though suppressions did not show this shift (Fig 6D and 6F, and Fig 7B, 7D and 7F).

**Complementary spatial distributions of small-scale and large-scale spectral-power activations and suppressions.** The red-blue regions in the spatial-probability-distribution plots are roughly reversed between small-scale (involving 1–4 sites) and large-scale (involving 10 + sites) activations/suppressions (Figs 3–7, parts A-F), suggesting that large-scale (involving 10

+ sites) activations and suppressions tended to cluster in the regions complementary to where small-scale activations and suppressions clustered. To statistically evaluate this spatial complementarity, we computed the vector similarity between the spatial distribution of focal activations/suppressions (involving only one activated/suppressed site) and the spatial distributions involving larger numbers of activated/suppressed sites. We first vectorized the spatial probability distribution for each number of activated/suppressed sites (per frequency band per condition per participant) by assigning the probability values (relative to the chance level of 1/60) at the 60 sites to a 60-dimensional vector. We then computed the angle, $A_n$, between the vector corresponding to the focal-activation (or focal-suppression) distribution, $\overrightarrow{V}_{focal}$, and the vector corresponding to the activation (or suppression) distribution involving $n$ sites, $\overrightarrow{V}_n$, that is,

$A_n = acos\left(\frac{\overrightarrow{V}_{focal} \cdot \overrightarrow{V}_n}{|\overrightarrow{V}_{focal}||\overrightarrow{V}_n|}\right)$. Note that, because the spatial probability distributions (computed relative to 1/60) have zero means, $A_n$ is equivalent to Pearson's correlation except for the $acos$ transform. For example, $A_n = 0°$ indicates that the distribution of $n$-site activations/suppressions is identical to the distribution of focal activation/suppression (up to a scalar multiplication factor), with $0° < A_n < 90°$ indicating that they are increasingly spatially dissimilar, $A_n = 90°$ indicating that they are orthogonal (unrelated), $A_n > 90°$ indicating that they are increasingly spatially complementary, and $A_n = 180°$ indicating a complete pattern reversal (red-blue reversal).

We note that even if the dynamics of activations and suppressions were stochastic, some focal activations/suppressions would occur by chance, which could be weakly clustered in arbitrary regions; then, multi-site activations/suppressions would be statistically less likely to be detected in those regions due to our use of percentile-based thresholds to define activations and suppressions. This statistical bias may raise $A_n$ for multi-site activations/suppressions to be slightly above 90° even when the underlying dynamics of activations and suppressions were stochastic. To discount this bias, we subtracted the $A_n$ values computed for the phase-scrambled controls (for increased reliability, three sets of control $A_n$ values were computed based on three independent versions of the phase-scrambled controls, then averaged) from those computed for the actual data. Nevertheless, the statistical bias considered here has little impact when evaluating the spatial distributions of activations/suppressions averaged across participants such as those shown in Figs 3–7 because any arbitrary (stochastic) clustering of focal activations/suppressions would have been inconsistent across participants so that they would have averaged out.

The $A$ values (corrected for statistical bias) are plotted for activations and suppressions as a function of the number of concurrently activated or suppressed sites ($n_{sites}$) for each condition in Figs 3–7 (see the line graphs on the right side). The thick lines with filled circles show the mean angles and the gray lines show the angles for the individual participants. The horizontal lines (red for activation and blue for suppression) just below and above 90° indicate statistically significant negative and positive deviations from 90° (Bonferroni-corrected, $\alpha = 0.05$, 2-tailed, with participant as the random effect). The switch from significantly below 90° to significantly above 90° indicates the transition of multi-site-activation/suppression distributions from being similar to the focal-activation/suppression distributions to being complementary to them. For all frequency bands across all conditions, small-scale multi-site-activation/suppression distributions for $n_{sites} \leq \sim 4$ remained similar to the focal-activation/suppression distributions ($A$ values significantly below 90°), whereas large-scale distributions for $n_{sites} \geq 10$ became complementary to the focal-activation/suppression distributions ($A$ values significantly above 90°) (Figs 3–7).

The lines with open circles show the *A* values computed based on participant-averaged spatial-distribution vectors (rather than computing the *A* values per participant then averaging them across participants). They generally swing more widely from approaching 0° to approaching 180°, exaggerating the distribution similarity and complementarity. This means that the spatial distributions of small-scale activations/suppressions (involving 1–4 concurrently activated/suppressed sites) and the complementary spatial distributions of large-scale activations/suppressions (involving 10+ concurrently activated/suppressed sites) were consistent across participants (thus some of the measurement noise averaged out when the *A* values were computed based on the participant-averaged spatial distributions). In contrast, if each participant had his/her unique complementary distributions of small-scale and large-scale activations/suppressions, the open-circle lines would have tended to converge to a flat line at $y = 90$°. This analysis confirms that large-scale (involving 10+ sites) spectral-power activations/suppressions clustered in regions that were complementary to where small-scale (involving 1–4 sites) spectral-power activations/suppressions clustered.

**Spatial distributions of large-scale spectral-power activations and suppressions.** Where did large-scale (involving 10+ sites) activations and suppressions cluster by frequency band?

For the $\theta$ band, large-scale activations and suppressions clustered in the posterior regions in the rest-with-eyes-closed condition (Fig 3A and 3B, right columns labeled "Large-scale"), whereas they clustered in the frontal and/or lateral-anterior regions in the rest-with-eyes-open-in-dark condition (Fig 3C and 3D). In the silent-nature-video condition, intermediate-to large-scale activations clustered in the posterior to lateral-anterior regions (Fig 3E), whereas large-scale suppressions clustered in the posterior regions (Fig 3F).

For the $\alpha$ band, large-scale activations and suppressions clustered in the posterior and medial-anterior regions in the rest-with-eyes-closed condition (Fig 4A and 4B). These patterns overall replicated in the two eyes-open conditions except that the medial-anterior cluster was weak in the rest-with-eyes-open-in-dark condition (Fig 4C and 4D) and both the posterior and medial-anterior clusters were weaker and the posterior cluster for activations was centrally shifted in the silent-nature-video condition (Fig 4E and 4F).

For the $\beta_1$ band, large-scale activations clustered in the posterior-to-lateral-posterior and lateral-anterior regions (Fig 5A), whereas large-scale suppressions clustered in the posterior regions (Fig 5B) in the rest-with-eyes-closed condition. Large-scale activations also clustered in the posterior and lateral-anterior regions in the rest-with-eyes-open-in-dark condition (Fig 5C), whereas large-scale suppressions primarily clustered in the lateral-anterior regions (Fig 5D). In the silent-nature-video condition, large-scale activations clustered in the posterior and lateral-posterior regions (Fig 5E), whereas large-scale suppressions clustered in the lateral-anterior regions (Fig 5F).

For the $\beta_2$ band, large-scale activations and suppressions clustered in the lateral-posterior and lateral-anterior regions (Fig 6A and 6B) in the rest-with-eyes-closed condition. In the two eyes-open conditions large-scale activations and suppressions both clustered in the lateral-anterior regions (Fig 6C–6F).

For the $\gamma$ band, the spatial distributions of large-scale activations and suppressions in all conditions (Fig 7A–7F) were essentially the same as those for the $\beta_2$ band in the two eyes-open conditions (Fig 6C–6F), clustering in the lateral-anterior regions (Fig 7A–7F).

## Contextual effects on the spatial distributions of spectral-power activations and suppressions

How did the spatial clustering of spectral-power activations and suppressions depend on context? The spatial scale of activations and suppressions (as indexed by the number of

concurrently activated/suppressed sites, $n_{sites}$) remained small or large sometimes, but rapidly expanded from small to large or contracted from large to small at other times (e.g., Fig 2A–2D). Are the distributions of activations and suppressions influenced by these temporal contexts? We examined the spatial distributions of spectral-power activations and suppressions under three representative contexts, (1) when the spatial extent of activations/suppressions remained within a given range of spatial scale (small, intermediate, or large), (2) when it fluctuated between the small-scale and intermediate-scale ranges, and (3) when it extensively expanded or contracted between the small-scale and large-scale ranges (Figs 8–12).

Both activations and suppressions yielded their characteristic average spatial distributions (i.e., averaged across all contexts; see Fig 3–7) when they fluctuated within the small-scale (1–4 sites) range or within the large-scale (10+ sites) range (see the thick horizontal arrows in Fig 8 [for the $\theta$ band], Fig 9 [for the $\alpha$ band], Fig 10 [for the $\beta_1$ band], Fig 11 [for the $\beta_2$ band], and Fig 12 [for the $\gamma$ band], part A [for activations] and part E [for suppressions], for each condition). In some cases, activations also yielded consistent spatial distributions when they fluctuated within the intermediate-scale (5–9 sites) range (see the thin horizontal arrows in Figs 8–12, part A, for each condition), though suppressions rarely yielded consistent distributions while fluctuating within the intermediate-scale range (see the thin horizontal arrows in Figs 8–12, part E, for each condition). When activations and suppressions fluctuated between the small-scale (1–4 sites) and intermediate-scale (5–9 sites) ranges (e.g., monotonically increasing from 2 sites to 8 sites, monotonically decreasing from 7 sites to 1 site, etc.), they tended to yield the characteristic small-scale distributions (see the curved arrows in Figs 8–12, part B [for activations] and part F [for suppressions], for each condition). In all these cases where activations and suppressions fluctuated within a specific spatial-scale range (small, intermediate, or large) or fluctuated between the small- and intermediate-scale ranges, we did not observe any directional asymmetry; that is, the spatial distributions of activations and suppressions were virtually identical in the contexts of their spatial expansions and contractions (not shown).

For extensive expansions and contractions of activations and suppressions, we observed some directional dependence so that the expansions from the small-scale (1–4 sites) to large-scale (10+ sites) range (e.g., monotonically increasing from 3 sites to 15 sites) and the contractions from the large-scale (10+ sites) to small-scale (1–4 sites) range (e.g., monotonically decreasing from 12 sites to 2 sites) are shown separately. Nevertheless, general characteristics are observed for both expansions and contractions. During the extensive spatial expansions and contractions of activations (see the unidirectional curved arrows in Figs 8–12, part C [for expansions] and part D [for contractions], for each condition), some consistent distributions were observed in the intermediate- and large-scale ranges including some of the characteristic large-scale distributions we saw in Figs 3–7, but no consistent distributions were observed in the small-scale range. During the extensive spatial expansions and contractions of suppressions (see the unidirectional curved arrows in Figs 8–12, parts G [for expansions] and H [for contractions], for each condition), no consistent spatial distributions were observed in any spatial scale.

Overall, the temporal-context analysis yielded some notable results. First, the characteristic spatial distributions of activations and suppressions shown in Figs 3–7 were especially consistent while the spatial scale of activations and suppressions remained within the small-scale or large-scale range, with the intermediate-scale distributions largely being transitional. Second, some of the characteristic large-scale spatial distributions of activations shown in Figs 3–7, but not suppressions, occurred during extensive expansions and contractions of activations. Third, none of the characteristic small-scale spatial distributions of activations or suppressions shown in Figs 3–7 occurred during extensive expansions or contractions of activations or suppressions. These results suggest that the characteristic small-scale and large-scale clusters of

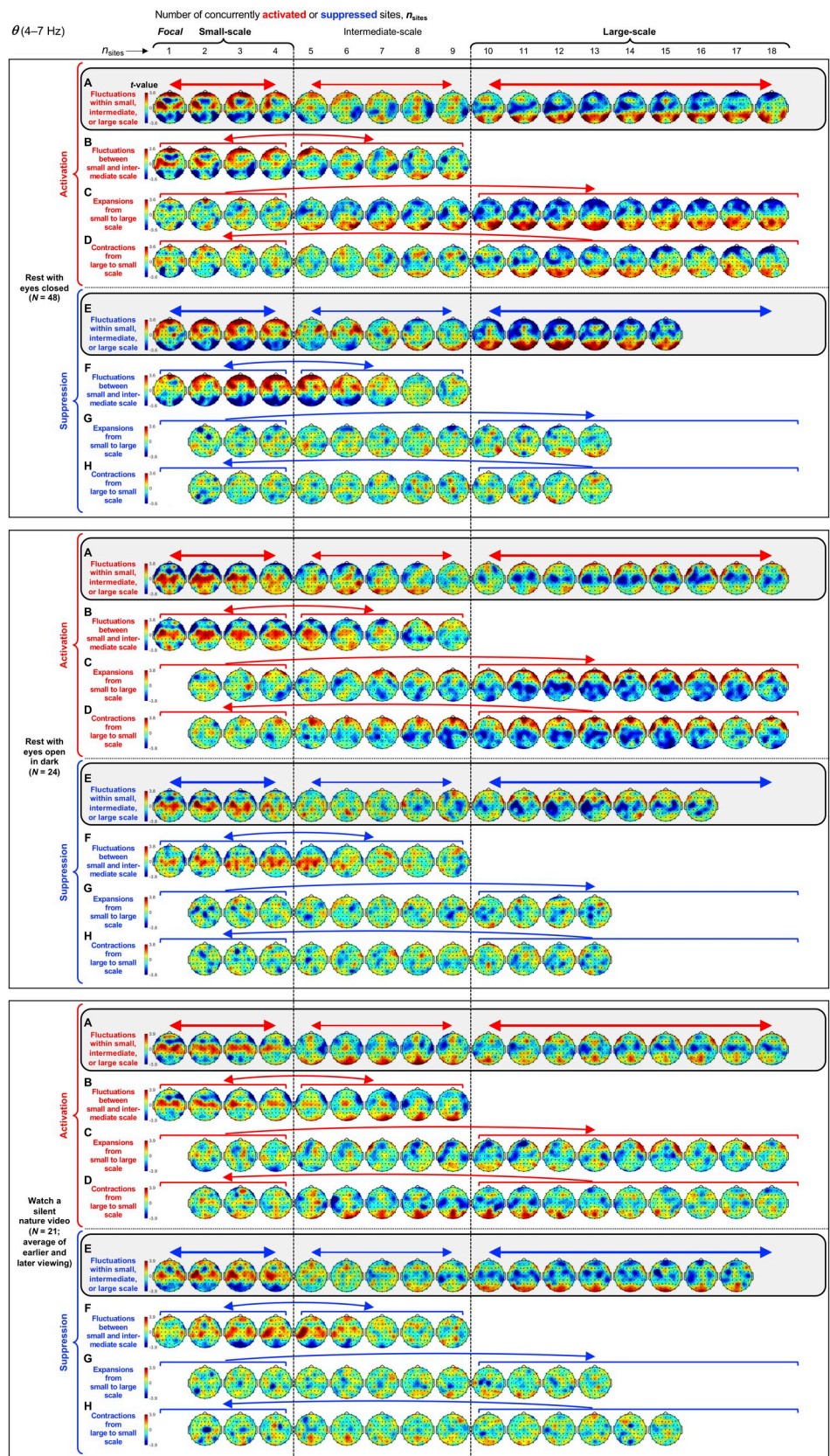

**Fig 8. Context effects on the spatial distributions of spectral-power activations and suppressions for the $\theta$ band (4–7 Hz).** As in Figs 3–7, activation and suppression are defined as spectral power in the top and bottom 8%, and the major row divisions correspond to the three behavioral conditions. **A and E per condition.** Spatial distributions of activations (**A**) and suppressions (**E**) when the number of concurrently activated or suppressed sites fluctuated within the small-scale (1–4 sites), intermediate-scale (5–9 sites), or large-scale (10+ sites) range (highlighted). **B and F per condition.** Spatial distributions of activations (**B**) and suppressions (**F**) when the number of concurrently activated or suppressed sites fluctuated between the small-scale (1–4 sites) and intermediate-scale (5–9 sites) ranges. **C and G per condition.** Spatial distributions of activations (**C**) and suppressions (**G**) when the number of concurrently activated or suppressed sites extensively expanded from the small-scale (1–4 sites) to large-scale (10+ sites) range. **D and H per condition.** Spatial distributions of activations (**D**) and suppressions (**H**) when the number of concurrently activated or suppressed sites extensively contracted from the large-scale (10+ sites) to small-scale (1–4 sites) range. All spatial distributions are shown in *t*-values as in Figs 3–7, parts A-F. Spatial-distribution plots are shown only for the cases where all participants contributed to the corresponding data. Compare these context-specific spatial distributions with the context-averaged distributions shown in Fig 3.

activations and suppressions, especially the small-scale clusters, emerge and dissolve while the spatial scale of activations and suppressions fluctuate within the corresponding range.

## Temporal correlations of the number of activated/suppressed sites between frequency bands

We have seen that the spatial scale of spectral-power activations and suppressions dynamically fluctuate between the characteristic small-scale and large-scale patterns that are frequency specific (Fig 2A–2D; Figs 3–7). Are the engagements of small- and large-scale networks independent or coordinated across frequency bands? To address this question, we computed the temporal correlation of $n_{site}$ (the number of concurrently activated or suppressed sites) between each pair of frequency bands, using Spearman's r, $r_{sp}$ (which is outlier resistant). To reduce any effects of spectral leakage (due to the wavelets' spectral-tuning widths; see Materials and methods), we computed the baseline-corrected $r_{sp}$ by subtracting from $r_{sp}$ the baseline $r_{sp}$ computed with phase-scrambled data (we actually computed three baseline $r_{sp}$ values based on three independent versions of phase-scrambled data, then averaged them).

Fig 13 presents the correlation matrices (symmetric about the diagonal) for activations (Fig 13A–13E) and suppressions (Fig 13K–13O) for each of the five behavioral conditions. The correlation patterns were generally similar for activations and suppressions. All average correlations were positive (lighter colors indicating stronger correlations) and statistically significant (based on $r_{sp}$ computed with Fisher-Z transformed $r_{sp}$ and baseline $r_{sp}$ values; $p < 0.05$, two-tailed, Bonferroni corrected), except for those indicated with the white "-" symbols. The temporal correlations of $n_{sites}$ were generally modest (averaging less than 0.5) and diminished with greater frequency separations; the cells farther away from the diagonal are darker. The $\theta$–$\beta_2$, $\theta$–$\gamma$, and $\alpha$–$\gamma$ correlations were especially weak as evident from the dark-colored cells in the upper-right and lower-left corners of all correlation matrices (Fig 13).

Interestingly, the correlation matrices for both activations and suppressions characteristically differed between the eyes-closed (rest-with-eyes-closed) and eyes-open (rest-with-eyes-open-in-dark and watch-a-silent-nature-video) conditions. Each of the eyes-closed matrices (Fig 13A and 13B for activations, and 13K and 13L for suppressions) is characterized by a large central region of lighter cells indicative of elevated correlations among the $\alpha$, $\beta_1$, and $\beta_2$ bands (highlighted with a large dashed central square), whereas each of the eyes-open matrices (Fig 13C–13E for activations, and 13M-13O for suppressions) is characterized by a pair of smaller square regions of lighter cells indicative of elevated correlations between the $\alpha$ and $\beta_1$ bands and between the $\beta_2$ and $\gamma$ bands (highlighted with two small dashed squares). To quantitatively evaluate the central-square and double-square patterns, we compared the $r_{sp}$ values (computed with Fisher-Z transformed $r_{sp}$ and baseline $r_{sp}$ values) among the cells labeled 1, 2, and 3. The

central-square pattern is distinguished by relatively higher $r_{sp}$ values in cells 1 and 2 with lower $r_{sp}$ values in cells 3. Although this pattern is not particularly strong in the eyes-closed conditions (Fig 13F and 13G for activations, and 13P–13Q for suppressions), the average $r_{sp}$ values were always lowest in cells 3. The reason why the $r_{sp}$ values were not substantially lower in cells 3 is that some of the participants yielded the double-square pattern corresponding to the eyes-open conditions with high correlations in cells 3 (see the thin lines showing the data from individual participants in Fig 13F, 13G and 13P, 13Q). The double-square pattern is distinguished by higher $r_{sp}$ values in cells 1 and 3 with lower $r_{sp}$ values in cells 2. This V-shaped pattern was consistently observed in the eyes-open conditions (Fig 13H, 13J for activations, and 13R, 13T for suppressions).

Thus, we obtained some evidence for coordinated engagements of the small-scale and large-scale synchronization networks between frequency bands (as temporal correlations of $n_{sites}$ between frequency bands). The coordination was generally modest (average $r_{sp}$ values being less than 0.5) and weaker between more distant frequency bands, being absent or nearly absent for the θ–$β_2$, θ–γ, and α–γ pairs. Notably, the pattern of coordination depended on the eyes being closed or open; when the eyes were closed the α, $β_1$, and $β_2$ bands tended to be somewhat jointly coordinated, but when the eyes were open the α-$β_1$ coordination tended to dissociate from the emerging $β_2$-γ coordination (with diminished correlations for the α-$β_2$ and $β_1$-$β_2$ pairs).

## Caveats

While the current analyses used a time-frequency decomposition approach that extracted sinusoidal components from EEG, oscillatory neural activities are not necessarily sinusoidal. As non-sinusoidal oscillations generate harmonics [e.g., 49, 50], some of our frequency-specific results may be contaminated by such artifactual harmonics. Nevertheless, it has been shown that macroscopic oscillatory activities from large neural populations such as those reflected in EEG tend to approximate sinusoidal waveforms due to extensive spatial averaging [51].

EEG spectral power reflects oscillatory as well as non-oscillatory neural activities, with the latter primarily reflected in the $1/f^\beta$ spectral background that may include contributions from random-walk type neuronal noise generated by the Ornstein-Uhlenbeck process [e.g., 52, 53], interplays between excitatory and inhibitory dynamics [e.g., 54], and other processes (see [55, 56] for a review). While our use of the temporal derivative of EEG substantially reduced the $1/f^\beta$ spectral background on the timescale of several seconds for the current EEG data (see Materials and methods), as β is known to fluctuate over time [e.g., 55], it is unclear the degree to which taking the temporal derivative continuously reduced the $1/f^\beta$ spectral background to highlight oscillatory activity. Thus, our results may reflect spatiotemporal fluctuations in the $1/f^\beta$ spectral background as well as spatiotemporal fluctuations in oscillatory synchronization and desynchronization.

Might the current results reflect non-neural artifacts? In particular, activities of the ocular and scalp muscles might have generated large spectral power at specific scalp sites. While muscle artifacts tend to occur in the γ band [57, 58], we observed characteristic small-scale and large-scale spatial distributions of spectral-power activations in all representative frequency bands (Figs 3–7). Further, the fact that the average durations of small- and large-scale activations were extended (relative to the phase-scrambled controls) by similar factors for all frequency bands (Fig 2) is inconsistent with the possibility that γ band activations may have uniquely reflected muscle artifacts. The strongest evidence we have against any substantive contributions of muscle artifacts to our results is that the characteristic spatial distributions of

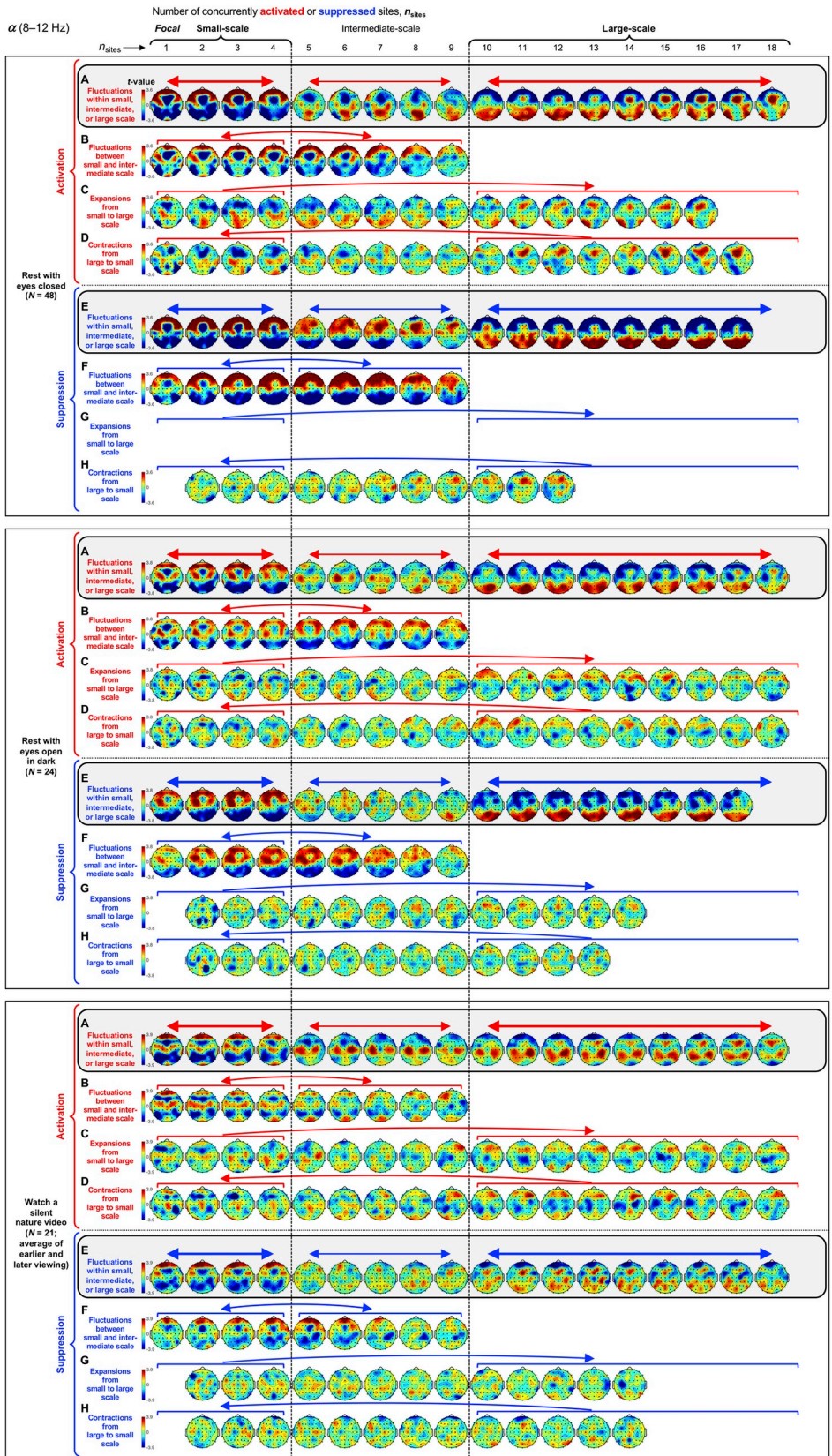

**Fig 9. Context effects on the spatial distributions of spectral-power activations and suppressions for the *α* band (8–12 Hz).** The formatting is the same as in Fig 8. No spatial-distribution plots are shown in **G** for the rest-with-eyes-closed condition because none of the relevant spatial distributions (for any value of $n_\text{sites}$) had data from all participants. Compare these context-specific spatial distributions with the context-averaged distributions shown in Fig 4.

activations and suppressions generally overlapped, especially for small-scale activations and suppressions and especially for *γ* band (Figs 3–7). Muscle artifacts would not generate consistent spatial distributions of spectral-power suppressions.

## General discussion

Oscillatory neural dynamics are prevalent in the brain [59–61], contributing to perceptual [17, 18, 62, 63], attentional [14, 21–23], memory [15], and cognitive [12, 13, 16, 19] processes (and probably to many other processes), likely through controlling information flow by adjusting phase relations within and across frequency bands [3, 10, 11, 22, 61]. While phase relations may be intricately coordinated across neural populations depending on functional demands, an impact of phase realignments would be particularly high (or low) when the oscillatory activities within the interacting neural populations are strongly synchronized (or desynchronized). We thus sought to uncover rules (if any) that govern the spatiotemporal dynamics of maximal and minimal spectral power, which may partly reflect the spatiotemporal dynamics of strong synchronization and desynchronization of cortical neural populations (see Introduction).

To this end, it was important to define maximal and minimal spectral power with appropriate thresholds. Instead of choosing the thresholds arbitrarily (e.g., top and bottom quartiles), we defined them empirically. To characterize how regions of maximal and minimal spectral power spontaneously expanded and contracted, isolated occurrences of maximal and minimal spectral power (occurring exclusively at a single site) may be informative. We thus chose the thresholds for which the occurrences of isolated maximal and minimal spectral power were most prevalent. Interestingly, the resultant thresholds were universal (virtually identical for different frequency bands across all behavioral conditions) and symmetric for maximal (top 8%) and minimal (bottom 8%) spectral power (Fig 1). This indicates that extreme spectral power at the top and bottom 8% are generally the most spatially isolated in the spatiotemporal dynamics of EEG spectral power. We thus operationally defined the top 8% spectral power as the state of "activation" of neural synchronization and the bottom 8% spectral power as the state of "suppression" of neural synchronization (while acknowledging the various concerns associated with relating EEG spectral power with neural synchronization; see Introduction and Caveats). Using these definitions, we obtained converging evidence suggesting that spectral-power activations and suppressions are organized into spatially segregated networks, relatively isolated small-scale networks comprising only 1–4 (2–7%) concurrently activated or suppressed sites and highly cooperative large-scale networks comprising 10–18 (17–30%) concurrently activated or suppressed sites.

First, instances of small-scale (involving 1–4 sites) and large-scale (involving 10+ sites), but not intermediate-scale (involving 5–9 sites), activations and suppressions were more prevalent and longer-lasting than expected from stochastic variations (Fig 2). This suggests that small-scale and large-scale activations and suppressions are actively maintained. Second, small-scale and large-scale (but not intermediate-scale) activations and suppressions consistently segregated in specific regions with generally overlapping distributions for activations and suppressions (Figs 3–7). This suggests that strong synchronizations and desynchronizations occur with elevated probabilities in spatially segregated small-scale and large-scale networks. Third, the characteristic spatial patterns of spectral-power activations and suppressions were

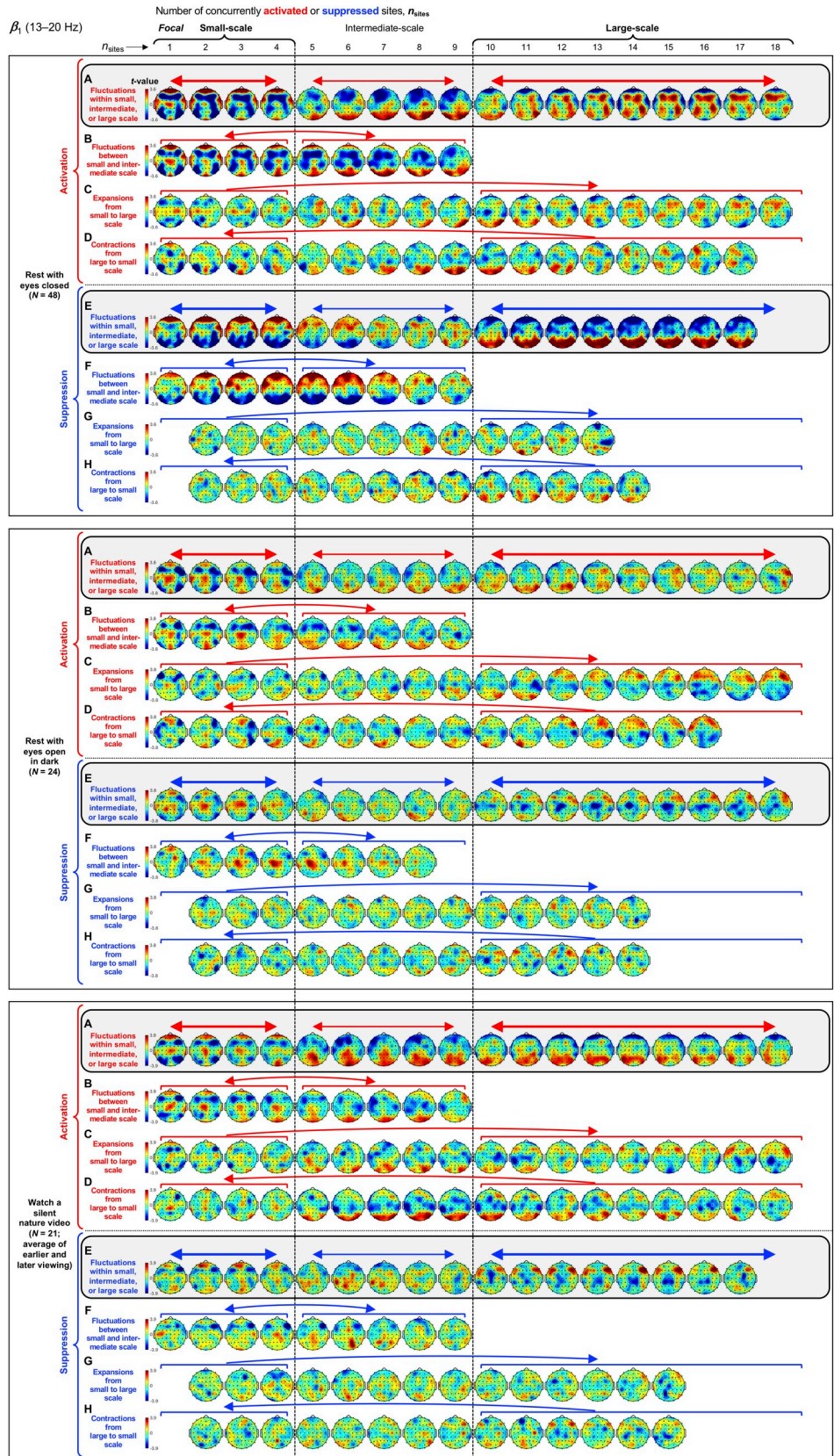

**Fig 10. Context effects on the spatial distributions of spectral-power activations and suppressions for the $\beta_1$ band (13–20 Hz).** The formatting is the same as in Figs 8 and 9. Compare these context-specific spatial distributions with the context-averaged distributions shown in Fig 5.

generally stable while the spatial extents of activations and suppressions fluctuated within the small-scale or large-scale (but not intermediate-scale) range (Figs 8–12). Some of the characteristic large-scale, but not small-scale, spatial patterns of activations were observed during extensive expansions and contractions of activations, but none of the characteristic patterns of suppressions were observed during extensive expansions or contractions of suppressions. These contextual dependences overall suggest that the characteristic small-scale and large-scale clusters of activations and suppressions primarily emerge and dissolve while activations and suppressions remain within the small-scale or large-scale range.

Taken together, these results suggest that the spatiotemporal dynamics of maximal and minimal spectral power, potentially indicative of the spatiotemporal dynamics of strong synchronizations and desynchronizations of oscillatory neural activities, are partly characterized by spatially segregated small-scale networks where regional populations are strongly synchronized or desynchronized in relative isolation and large-scale networks where many regional populations are strongly synchronized or desynchronized concurrently in a highly cooperative manner. These small-scale and large-scale networks of concurrently synchronized/desynchronized neural populations may confer an overarching dynamic structure that may constrain the impact of task-specific realignments of phase relations within and across frequency bands and brain regions. Future research may investigate how phase-relation realignments are coordinated with the status (strongly synchronized or desynchronized) of these small- and large-scale synchronization networks to optimize information processing.

Our results also suggest that these small-scale and large-scale networks are frequency specific (Figs 3–7 and Figs 8–12). Speculating on the potential relationships between the frequency-specific networks identified here and the extensive literature on task-dependent oscillatory activities involving different frequency bands is beyond the scope of the current discussion. Instead, we summarize some notable aspects of the frequency specificity of the small- and large-scale synchronization/desynchronization networks. First, in the eyes-closed condition, the small-scale networks for the $\theta$, $\alpha$, $\beta_1$, and $\beta_2$ bands all included the frontal region, while the large-scale networks systematically shifted from including primarily posterior regions for the $\theta$ band, posterior and medial-anterior regions for the $\alpha$ band, posterior and lateral-anterior regions for the $\beta_1$ band, primarily lateral-anterior regions for the $\beta_2$ band, and consistently lateral-anterior regions for the $\gamma$ band (Figs 3–7, parts A and B). Second, regardless of behavioral condition, the small-scale networks were consistently localized in the medial-central-posterior regions and the large-scale networks were localized in the lateral-anterior regions for the $\gamma$ band (Fig 7). Third, the small- and large-scale networks for the $\beta_2$ band clustered nearly identically to those for the $\gamma$ band in the eyes-open conditions regardless of visual stimulation (eyes open in dark and watching a nature video) (compare Fig 6C–6F with Fig 7A–7F). Further, the inter-frequency temporal coordination of small- and large-scale networks (in terms of the inter-frequency temporal correlations of $n_{\text{sites}}$ for both activations and suppressions) were also uniquely elevated between the $\beta_2$ and $\gamma$ bands in the eyes-open conditions (Fig 13, cells 3 on the right side labeled "Eyes open"). These results suggest that close spatiotemporal coordination of synchronization and desynchronization between the $\beta_2$ and $\gamma$ bands is uniquely associated with the state of the eyes being open (regardless of visual stimulation). In general, the engagements of small- and large-scale synchronization networks

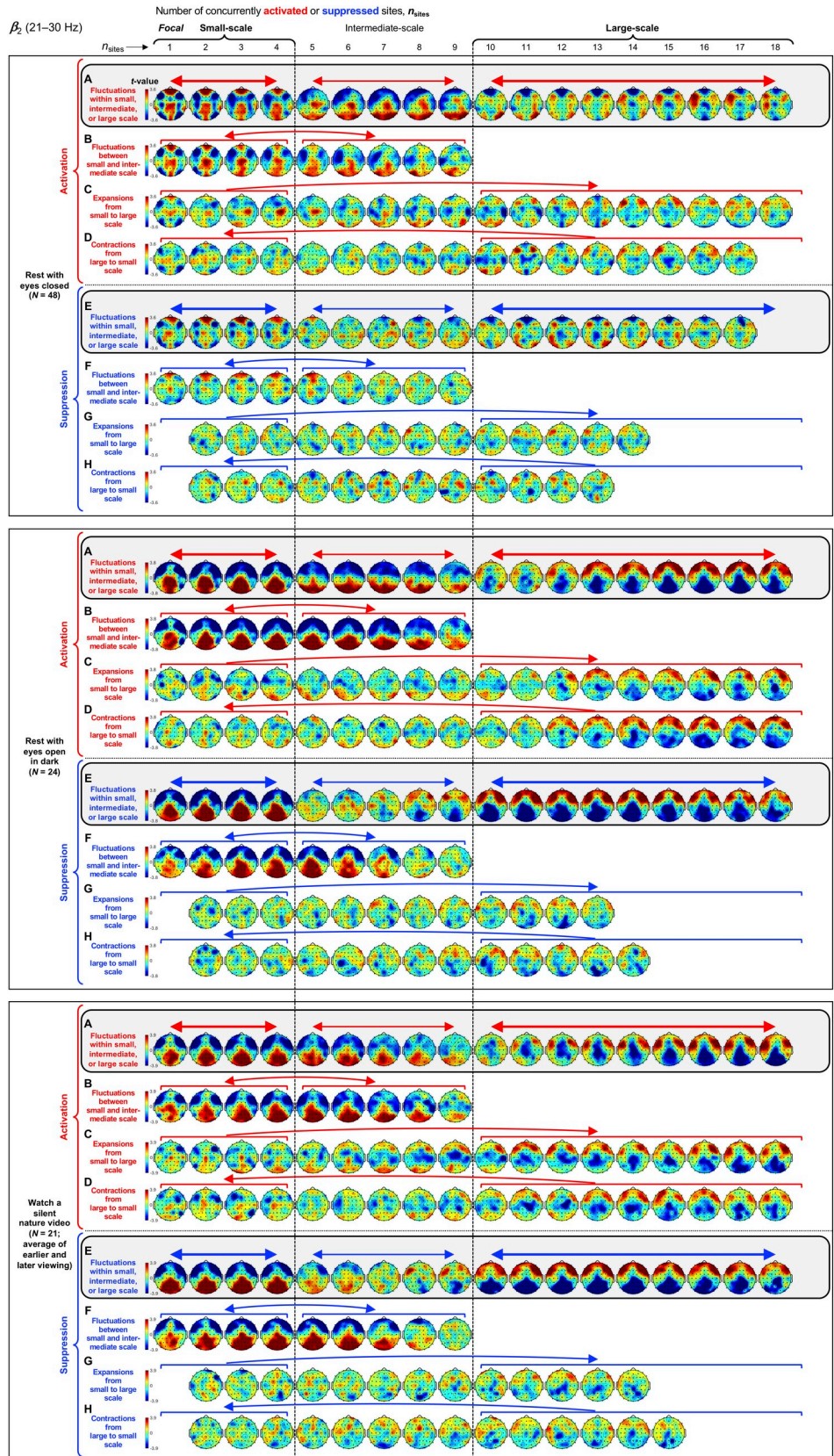

**Fig 11. Context effects on the spatial distributions of spectral-power activations and suppressions for the $\beta_2$ band (21–30 Hz).** The formatting is the same as in Figs 8–10. Compare these context-specific spatial distributions with the context-averaged distributions shown in Fig 6.

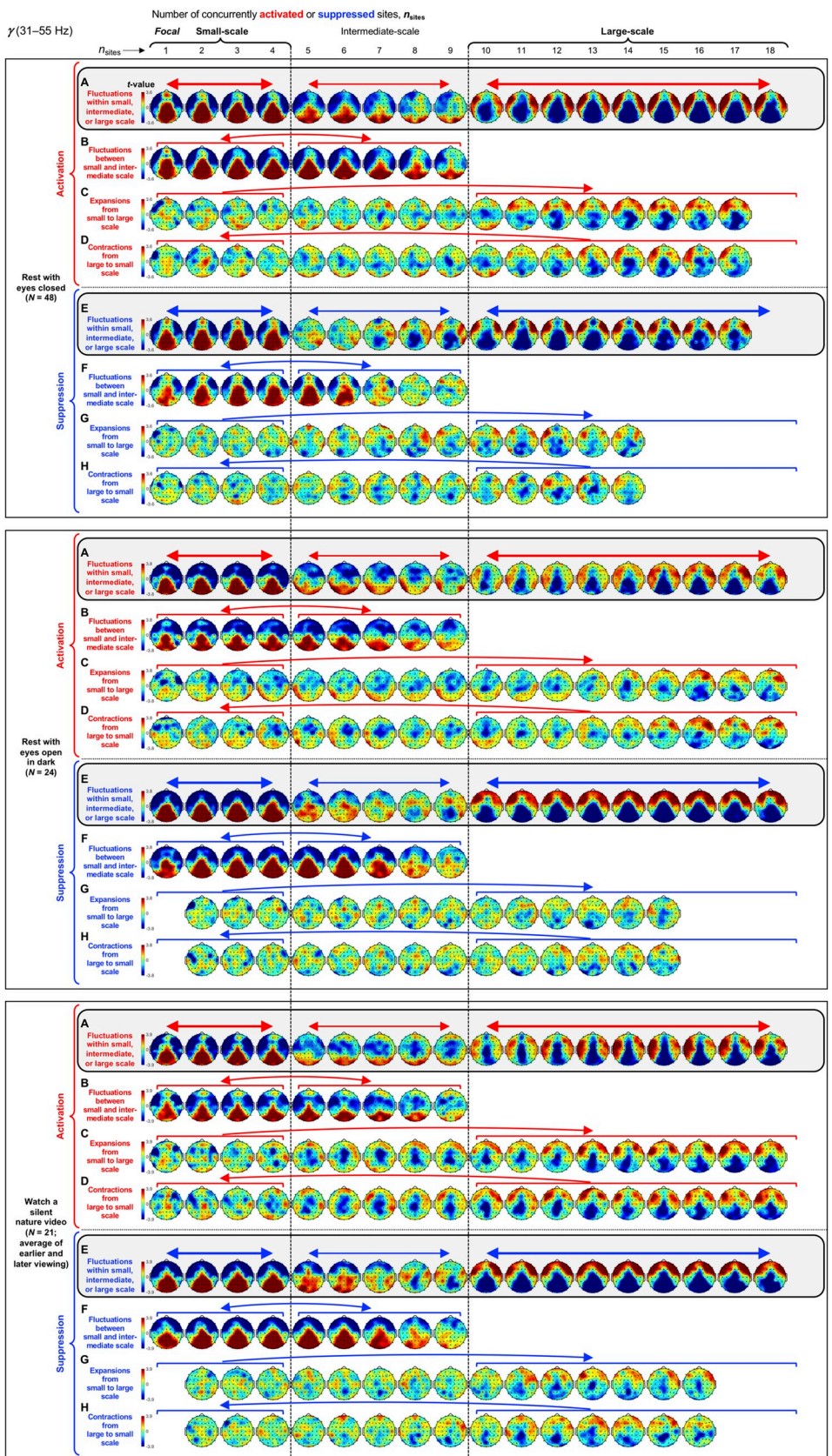

**Fig 12. Context effects on the spatial distributions of spectral-power activations and suppressions for the $\gamma$ band (31–55 Hz).** The formatting is the same as in Figs 8–11. Compare these context-specific spatial distributions with the context-averaged distributions shown in Fig 7.

were only modestly coordinated ($r_{sp} < 0.5$) across frequency bands, and virtually uncoordinated between distant frequency bands (Fig 13).

To conclude, many studies have examined the spatial structures of temporal associations among spectral amplitudes and phases within and across frequency bands, typically analyzing structures of correlation matrices, to identify (static, time-averaged) spatial networks and their

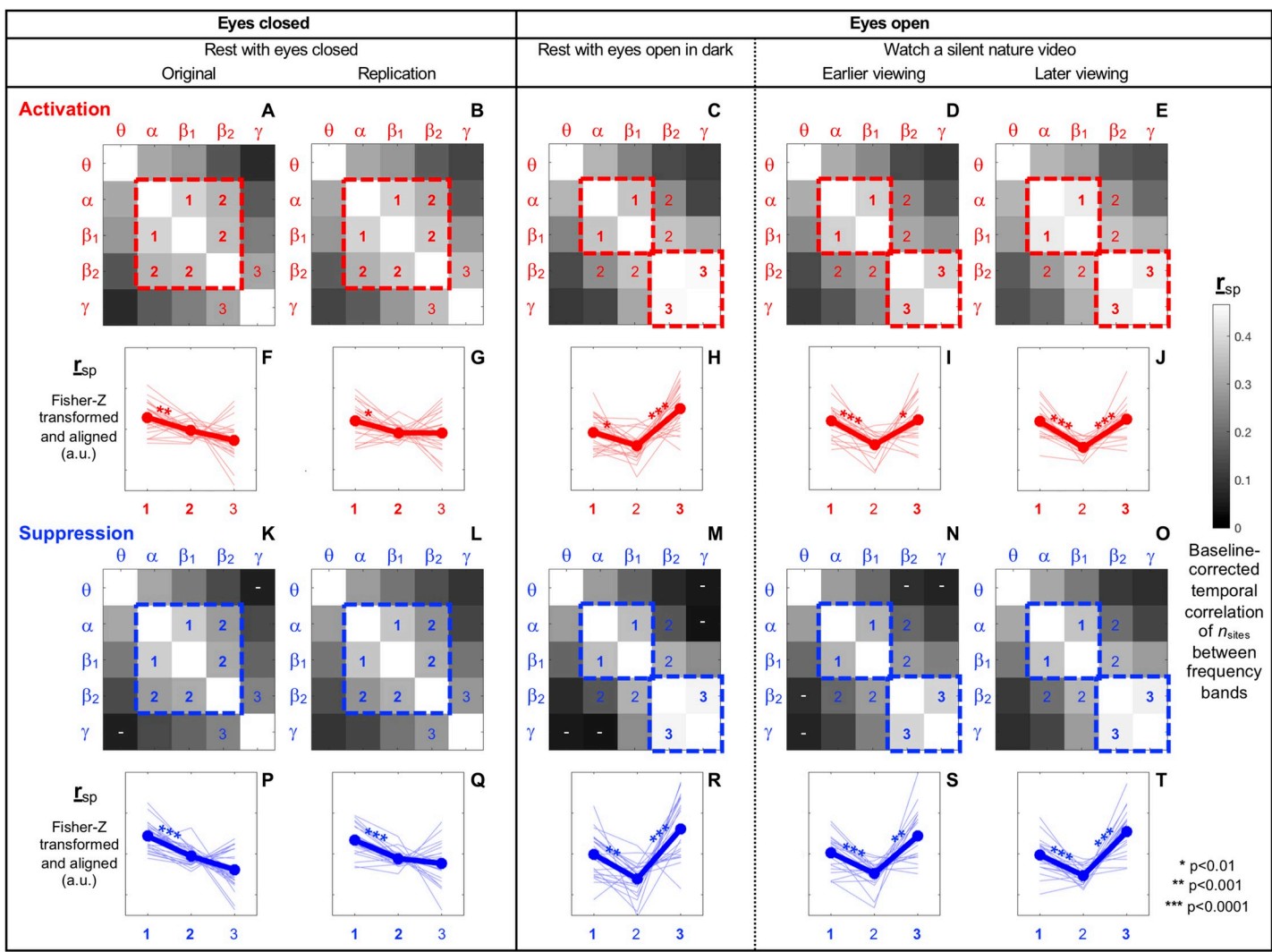

**Fig 13. Temporal correlations of $n_{sites}$ (the number of concurrently activated/suppressed sites) between frequency bands.** The correlations were computed as baseline-corrected Spearman's r, $r_{sp}$ ($r_{sp}$ minus baseline-$r_{sp}$ computed based on phase-scrambled data) per frequency-band pair per condition per participant (see main text). **A-E.** Correlation matrices of $n_{sites}$ (symmetric about the diagonal) for spectral-power activations for the five behavioral conditions. Larger $r_{sp}$ values are indicated with lighter colors (all positive). All $r_{sp}$ values were statistically significant at $p<0.05$ (two-tailed, Bonferroni corrected). **K-O.** The same as A-E, but for spectral-power suppressions. The "-" symbols indicate statistically non-significant $r_{sp}$ values. **F-J.** Line graphs comparing $r_{sp}$ values among the cells labeled 1, 2, and 3 in the correlation matrices A-E (see main text). The $r_{sp}$ values here were computed with Fisher-Z transformed $r_{sp}$ and baseline- $r_{sp}$ values to be appropriate for statistical comparisons. The thick lines with closed circles indicate the means and the thin lines indicate the data from individual participants (all lines have been aligned at their respective means to highlight the inter-participant variability in cell dependence). **P-T.** The same as F-J but for spectral-power suppressions.

connectivity that may be associated with specific behavioral functions and mental states. What was unique about the current study was to investigate general rules governing the spatiotemporal dynamics of strongly synchronized and desynchronized neural populations by tracking the dynamics of spontaneously emerging and dissipating clusters of maximal and minimal spectral power. In this way, we were able to obtain converging evidence suggesting that prolonged periods (compared with stochastic dynamics) of strong synchronization and desynchronization occur in small-scale and large-scale networks that are spatially segregated and frequency specific, macroscopically segregating the relatively independent and highly cooperative oscillatory processes.

## Author Contributions

**Conceptualization:** Satoru Suzuki.

**Data curation:** Melisa Menceloglu.

**Formal analysis:** Satoru Suzuki.

**Funding acquisition:** Melisa Menceloglu, Marcia Grabowecky.

**Investigation:** Melisa Menceloglu, Satoru Suzuki.

**Methodology:** Satoru Suzuki.

**Project administration:** Marcia Grabowecky, Satoru Suzuki.

**Resources:** Melisa Menceloglu, Marcia Grabowecky, Satoru Suzuki.

**Software:** Melisa Menceloglu, Satoru Suzuki.

**Supervision:** Marcia Grabowecky, Satoru Suzuki.

**Validation:** Satoru Suzuki.

**Visualization:** Satoru Suzuki.

**Writing – original draft:** Satoru Suzuki.

**Writing – review & editing:** Melisa Menceloglu, Marcia Grabowecky, Satoru Suzuki.

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
