## [Decision Letter · Decision Letter 0]

2 Mar 2021

PONE-D-21-01871

Spatiotemporal dynamics of maximal and minimal EEG spectral power

PLOS ONE

Dear Dr. Suzuki,

Thank you for submitting your manuscript to PLOS ONE. After careful consideration, we feel that it has merit but does not fully meet PLOS ONE’s publication criteria as it currently stands. Therefore, we invite you to submit a revised version of the manuscript that addresses all the points raised during the review process.

Although both Reviewers, as well as the Editor, acknowledge the interest and relevance of the study, many issues need to be resolved. A particular attention has to be paid on the comments/criticisms from the Reviewer #1 who raised serious issues, including the scientific/conceptual background of the study, the pertinence of the methodologies as well as the figures design and their appropriateness to illustrate the central findings. The help of a third Reviewer will be probably necessary.

We look forward to receiving your revised manuscript.

Kind regards,

Stéphane Charpier

Academic Editor

PLOS ONE

Journal Requirements:

Reviewers' comments:

Reviewer's Responses to Questions

**Comments to the Author**

1. Is the manuscript technically sound, and do the data support the conclusions?

Reviewer #1: No

Reviewer #2: Yes

2. Has the statistical analysis been performed appropriately and rigorously? 

Reviewer #1: I Don't Know

Reviewer #2: Yes

3. Have the authors made all data underlying the findings in their manuscript fully available?

Reviewer #1: Yes

Reviewer #2: Yes

4. Is the manuscript presented in an intelligible fashion and written in standard English?

Reviewer #1: No

Reviewer #2: Yes

5. Review Comments to the Author

Reviewer #1: The authors report on a series of analyses investigating spatial-temporal dynamics of spontaneous frequency power fluctuations in EEG, recorded under resting-state conditions.

In the broad sense, the manuscript suffers from the fact that the methods are introduced throughout the (discussion of) the results. The decisions on which methods to use, how, and to what end, should instead be fully discussed in the introduction and method section. A related problem is that the manuscript is replete with qualitative statements and post-hoc decisions such as which thresholds to use, and how many channels to consider “intermediate-scale activations”. Furthermore, as these are introduced during a discussion of the results, a situation is created where it is impossible to discern a-priori predictions from ‘cherry-picking’ result. Therefore, all decisions regarding operationalization of variables, categories, parameters, etc. have to be declared in the methods section in a quantitative a-priori manner, or clear arguments given why this would not be possible. Qualitative statements, (e.g.: “moderately distributed”, large number of neurons”, “strongly synchronize” “virtually eliminated” “maximum sensitivity (non-numeric), “generally shorter”, “especially high spectral power”) should be minimized as much as possible, at least until the discussion of the results.

The general research question remains unclear as the background is not reviewed and explained in any degree, and the concepts are introduced in a unnecessarily complicated and over-generalized manner. For example, while the following two sentences from the introduction give 25 references, they do not provide any details that would clarify the scientific background, methodologies or past empirical findings, while lumping together different kinds of neuronal connectivity metrics, behavior and mental states: “Those studies characterized network features such as modules, hubs, and motifs based on the spatial organizations of amplitude relations, phase relations, and information transfer [1–8; see 9 for a review]. This line of investigation has revealed frequency-specific hierarchical networks (including networks and sub-networks) whose structures and activities correlate with a variety of behavioral functions [10–22] and mental states [23–26].”

The manuscript present results that remain largely descriptive, without much synthesis or hypothesis testing. Although the completeness of the presentation of the results is appreciated, the figures leave the reader uncertain about which results are the most relevant. Furthermore, while the results depend on cognitive state and the conditions under which they were recorded, as well as the frequency dependent, no explanation is given.

The manuscript also misses a discussion of the physiological reality and relevance of the approach and the findings. E.g. the use of “suppression” and “activation”, and purported physiological mechanisms, are not substantiated by references to existing literature or empirical data. E.g. the following sentence seems to erroneously assume that EEG oscillations are the result of “oscillating neurons”, and that these represents an “activation” state: “…we respectively refer to as "activation" and "suppression," reflective of strong within-sub-population synchronization (a large number of neurons synchronously oscillating) and desynchronization (only a small number of neurons synchronously oscillating or a sizable number of neurons incoherently oscillating).”

Together, I am of the opinion that the manuscript lack the high degree of scientific rigor necessary for publication.

Reviewer #2: The authors performed the spectral analysis of multichannel EEG signals recorded in healthy persons during the resting state. The results of this study and the way they are presented are very interesting and original.

However, the assumption that the power spectrum is a measure of brain synchronization is not correct. The phase synchronization methods like Phase Locking Value or Phase Lag Index are used most often for this purpose. Moreover, the authors having the sixty-channel scalp EEG data, were able to perform the source-level spectral analysis.

There are several incorrect citations in this paper. The expression: “see citations above” should be replaced by appropriate numbers of references. Moreover, the authors should not cite their paper that are under review and it has not yet been published (reference 27). More citations are needed in the Introduction section (the last paragraph on the page 3 and the last paragraph of the Introduction section on the page 4). The results described in the Discussion section should be compared with the results of other authors.

The term ‘1 non-binary’ in the first phrase of ‘Participants’ section is not clear. The symbols M and SD can be replaced by ‘mean age: X ± Y years’.

There are several phrases which should be explained better.

In ‘EEG recording and preprocessing’ section: “To reduce effects of volume conduction (to approximately within adjacent sites; e.g., [32] ???), to virtually eliminate the effects of reference electrode choice (verified ??? ), as well as to facilitate data-driven determinations of EEG current sources ??? ”;

In ‘EEG analysis’ section several expressions are unclear: “macroscopically estimating the underlying electrical currents”; “without having to estimate and discount them in sliding-time windows”; “stochastically (i.e., unpredictably in a memory-free manner)”.

In ‘Spatial distributions of spectral-power activations and suppressions’ section: “capped at 1/nsites – 1/60”; “floored at -1 (100% below the chance level)”.

The bottom row of Figure 2 with remaining four conditions is redundant because results are similar to that for the first condition (rest with eyes closed). Whereas, two panels in the upper row could be divided in two separated figures.

The descriptions of Figures, especially Figures 1 and 2, are too long. It should be moved to the main text. On the page 24 there is an error in the numbering of cited figures (Figures 8-12 should be replaced by Figures 3-7 in the second paragraph).

6. PLOS authors have the option to publish the peer review history of their article (what does this mean?). If published, this will include your full peer review and any attached files.

Reviewer #1: No

Reviewer #2: No

---

## [Author Response · Author response to Decision Letter 0]

14 May 2021

Please see the files uploaded under "Cover letter" and "Response to reviewers."

---

## [Editor Report · Decision Letter 1]

14 Jun 2021

Spatiotemporal dynamics of maximal and minimal EEG spectral power

PONE-D-21-01871R1

Dear Dr. Suzuki,

We’re pleased to inform you that your manuscript has been judged scientifically suitable for publication and will be formally accepted for publication once it meets all outstanding technical requirements.

Kind regards,

Stéphane Charpier

Academic Editor

PLOS ONE
---

## [Editor Report · Acceptance letter]

7 Jul 2021

PONE-D-21-01871R1 

Spatiotemporal dynamics of maximal and minimal EEG spectral power 

Dear Dr. Suzuki:

I'm pleased to inform you that your manuscript has been deemed suitable for publication in PLOS ONE. Congratulations! Your manuscript is now with our production department. 

Kind regards, 

on behalf of

Pr. Stéphane Charpier 

Academic Editor

PLOS ONE